# Harnessing Input-adaptive Inference for Efficient Vision-and-Language Navigation

## Abstract

An emerging paradigm in vision-and-language navigation (VLN) is the use of history-aware multi-modal transformer models. Given a language instruction, these models take observation and history as input and predict the most appropriate action for an agent. While employing these models has significantly improved performance, the *scale* of these models can be a bottleneck in practical settings where computational resources are limited (e.g., in robots). In this work, we present a novel input-adaptive navigation method for efficient VLN. We first characterize the overthinking problem in VLN and show that none of the existing input-adaptive mechanisms successfully reduce overthinking without causing significant performance degradation. Our method addresses this problem by developing three adaptive algorithms deployed at different levels: (1) We develop an adaptive approach that improves *spatial* efficiency; we only process a subset of panoramic views at each observation of an agent. (2) We also achieve *model-level* efficiency by developing adaptive thresholding for the early-exit method we employ, based on the importance of each view in navigation. (3) To achieve *temporal* efficiency, we design a caching mechanism to avoid processing views that an agent has seen before. In evaluations with six VLN benchmark tasks, we demonstrate over a $2\times$ reduction in computation across two off-the-shelf VLN agents.

## 1 Introduction

Progress in vision-and-language navigation (VLN) has been enabled by larger models (Hao et al., 2020; Chen et al., 2021; Moudgil et al., 2021; Guhur et al., 2021; Hong et al., 2021) trained on increasingly large datasets (Wang et al., 2023). These models can process and interpret complex data, enabling them to understand and act upon natural language instructions within visual environments. Despite the success, there is a growing concern about their computational demands. The need for substantial computational power poses a significant challenge for deployment in resource-constrained settings, such as robots, where low-power consumption increasingly becomes critical.

A potential solution to addressing these computational demands is *input-adaptive inference*. The main idea is to reduce *overthinking* (Kaya et al., 2019): as shallow networks are sufficient for the majority of samples to make decisions, e.g., class predictions, input-adaptive methods (Huang et al., 2018; Liu et al., 2020; Xin et al., 2020; Tang et al., 2023) stop forwarding preemptively during inference and return intermediate outputs when the internal decisions of a model converge. During inference, they demonstrate up to 50% computational savings while preserving model performance.

In this work, we study the overthinking problem in a new domain—vision-and-language navigation—and address this issue by proposing a novel input-adaptive navigation method. Prior work has focused on tasks where inputs are processed independently of one another, e.g., in classification. But they do not account for sequential decision tasks like navigation, where a model makes several decisions over time based on inputs with *spatio-temporal dependencies*. In consequence, existing methods are designed to reduce overthinking within a model but do not consider the *cognitive overload* caused by the visual inputs the model needs to process in navigation. Although these models are not yet widely deployed in real-world navigation tasks, it is still crucial to evaluate whether our efficiency gains are robust to common visual corruptions.

**Contributions.** We *first* characterize the overthinking problem in vision-and-language navigation. In our evaluation with two popular agents, HAMT (Chen et al., 2021) and DUET (Chen et al., 2022),

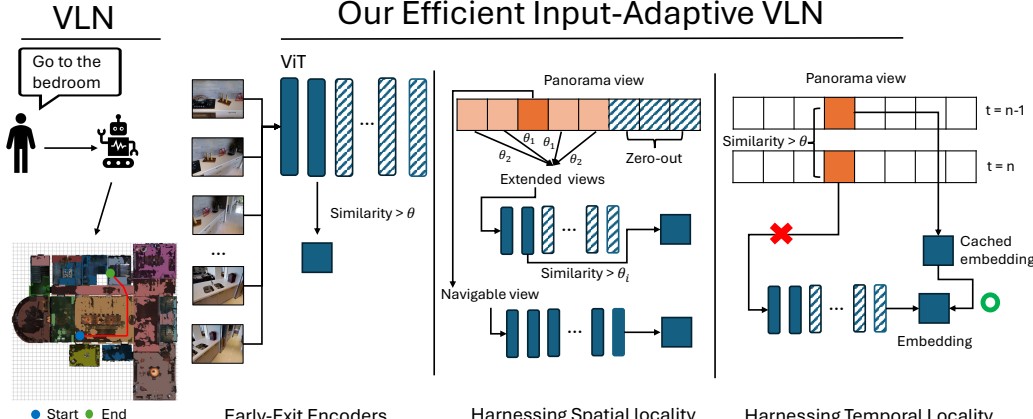

Figure 1: **Overview of our efficient input-adaptive navigation method.** We illustrate on the left an agent navigating a visual environment upon a natural language instruction. On the right, we provide a high-level overview of the three input-adaptive mechanisms we propose at different levels.

we find that ∼99.5% of computations are spent in Transformer-based encoders to compute visual encodings. We also show that addressing overthinking within these visual encoders is ineffective in providing computational savings. Even with our best effort to apply the existing input-adaptive inference method, MuE (Tang et al., 2023), we demonstrate that this approach results in unsuccessful and inaccurate navigation decisions. This, in turn, increases both the time it takes for an agent to reach the target location and the overall computations, while lowering the navigation success.

*Second*, to address this issue and achieve computational efficiency, we present a novel input-adaptive navigation method (shown in Figure 1). We not only minimize overthinking within visual encoders, as in prior approaches, but also reduce overthinking caused by *cognitive overload* during navigation. Specifically, we focus on exploiting the spatio-temporal localities unique to VLN tasks: (1) The spatial locality: In a panorama, we find that navigable views and a few neighboring views are critical for successful navigation. We design a weighting mechanism that significantly reduces the number of views the encoder should process. (2) The temporal locality: We also find that an agent encounters identical or nearly identical views in consecutive navigation steps. We design a locality-sensitive hashing algorithm to avoid computing these matching views during navigation. (3) At the model level, we develop an algorithm for dynamically adapting the thresholds for an existing early-exit method based on the locality to further reduce computations.

*Third*, we comprehensively evaluate our input-adaptive navigation method on 6 VLN benchmarks across two popular agents. Our method achieves significant computational savings of up to 60%, with a maximum performance degradation of 11.5% in SR. In contrast, baseline methods experience up to 33.6% performance loss and fail to reduce any computations. Our ablation study also shows how a practitioner can configure our method for their navigation environments and the factors that our method does not rely on. Moreover, we examine the robustness of our method to natural visual corruptions that may occur in navigation environments (such as lighting changes). We show that while both the baseline and our method show a slight increase in the computations, our approach loses 7–10% more performance. We hope our results will inspire future research on developing efficient navigation methods and their deployment in real-world VLN settings.

## 2 RELATED WORK

**Vision-and-language navigation (VLN).** Research in this area has been supported by the development of high-quality simulators such as Matterport3D (Chang et al., 2017) which we heavily leverage in our work. Agents developed towards this challenging problem have ranged from earlier recurrent models (Anderson et al., 2018; Fried et al., 2018) to more recent transformer-based models (Hong et al., 2021; Chen et al., 2021; 2022; Wang et al., 2023; Kamath et al., 2023). While the recent agents achieve superior visual-language alignment and improved performance, their increased complexity leads to higher computational costs during inference. One reason for this can be attributed to the transformer's quadratic complexity in the length of input tokens (Vaswani, 2017). In addition, they

operate in environments with panoramic observation and action space, often consisting of multiple single-view images (Fried et al., 2018). Our work is the first providing a tunable trade-off between computational demands and accuracy.

**Input-adaptive mechanisms for computational efficiency.** Prior work introduces two distinct mechanisms for input-adaptive inference: adaptive neural networks (AdNNs) and multi-exit architectures. AdNNs (Wang et al., 2018; Figurnov et al., 2017) are designed to dynamically skip certain blocks of the model to save the computations during inference. In contrast, multi-exit architectures (Teerapittayanon et al., 2016; Huang et al., 2018; Kaya et al., 2019; Xin et al., 2020) introduce an additional component to the model, such as classifiers attached to each internal layer (early-exits), allowing the model to preemptively stop running forwards once stopping criteria are met. Both mechanisms demonstrate computational savings while minimizing performance loss in classification tasks (e.g., a 50% reduction in computation at a utility loss of $\sim$10%). Our method employs multi-exit architectures because AdNNs are constrained to ResNet-like architectures. Most multi-exit architectures are developed for classification tasks and are *not* compatible with VLN, where an agent utilizes visual and/or language representations generated from encoders. The closest work by Tang et al. (2023) developed an adaptation (MuE) to Transformer-based encoders. But despite our best effort, MuE does not provide any computational savings in VLN tasks (shown in Sec 4.1). A separate line of research explores methods for compressing models, such as quantization and pruning. These methods are orthogonal to our study and can be applied in conjunction with our method (see Appendix F).

## 3 EXPERIMENTAL SETUP

**Datasets.** Following prior work, we evaluate our method on six different datasets: Room-to-Room (R2R) (Anderson et al., 2018), R2R-Back (Chen et al., 2021), R2R-Last (Chen et al., 2021), REVERIE (Qi et al., 2020), CVDN (Thomason et al., 2020), and SOON (Zhu et al., 2021).

**VLN Agents.** We consider two off-the-shelf VLN agents: HAMT (Chen et al., 2021) and DUET (Chen et al., 2022). HAMT has four components: (1) the Vision Transformer (ViT) (Dosovitskiy et al., 2021) that compute visual representations from panorama view captured at the current navigation step, (2) BERT (Devlin et al., 2019) that returns language representations from human instructions, (3) the Hierarchical ViT that progressively encode temporal dynamics across panoramas in the history into representations, (4) the Cross-modal Transformer that captures multi-modal relationships between the three representations, and predicts the most appropriate action. DUET consists of five components but operates differently from HAMT. (1) the ViT extracts visual representations and object features (e.g., bounding boxes and labels) from panorama views, (2) and uses the panorama encoder on them for topological mapping. For global action planning, (3) the text encoder generates language representations of human language instructions. Next, (4) the Coarse-scale Cross-modal Encoder determines the next action within a global action space, while (5) the Fine-scale Cross-modal Encoder selects actions within a local action space. The final action is then made by fusing the both.

**Evaluation metrics.** To evaluate how successful an agent's navigation is, we employ four metrics from prior work (Chen et al., 2021; Krantz & Lee, 2022): (1) Trajectory length (TL): the path length an agent navigated in meters. (2) Oracle Success rate (OSR): the ratio of trajectories where at least one viewpoint along the agent's path can see the target object within a 3 meter range. (3) Success rate (SR): the ratio of trajectories where an agent's final position is within 3 meters of the target. (4) Success rate normalized by inverse path length (SPL): the SR normalized by the ratio between the shortest path's length and the predicted path's length. We compute the total giga floating-point operations per second (GFLOPs) an agent requires to navigate to measure the computational savings our method offers. We also measure the wall-time (in seconds) it takes for an agent to navigate to its final location. But, we prioritize GFLOPs over wall-time because the latter depends on the hardware accelerators and the software implementation. Please refer to Appendix for more details.

## 4 INPUT-ADAPTIVE EFFICIENT VISION-AND-LANGUAGE NAVIGATION

### 4.1 CHARACTERIZING THE OVERTHINKING PROBLEM IN VLN

**Computational bottleneck.** The first step in designing an efficient input-adaptive mechanism is to understand the computational bottleneck of an agent during navigation. Because no prior work has

studied which component consumes the most computational resources, we take a representative VLN agent (HAMT) and analyze the computational demands of each of the five components. We use the pre-trained VLN agent, provided by the original study through the open-source repository. We run the agent on the validation (unseen) set of R2R and measure the average GFLOPs per trajectory.

Figure 2 summarizes our result. The component that requires the least computations is BERT (0.04%). This is because the agent uses BERT only once at the beginning of navigation to compute the language representation of a human instruction. We show that 99.5% of the computations are spent on the ViT, in computing visual representations from panorama views. Note that a single panorama view is composed of 36 views. This means that each time the agent makes a navigation decision, the ViT should process $36\times$ times more inputs than the remaining components (H-ViT and CMT). The computations these two models use depend on the input dimension and the number of Transformer layers, but they only account for 0.46% of the total computations the agent uses. We therefore focus on the ViT.

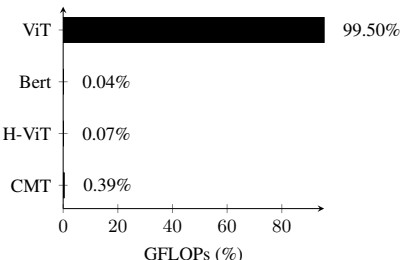

Figure 2: **Component-wise computational demands.** We run HAMT on the validation (unseen) set of R2R.

**Existing mechanisms are ineffective in saving computations in VLN.** Next, we examine whether existing input-adaptive inference methods are effective in providing computational savings in VLN. We find that most approaches discussed in Sec 2 are incompatible with VLN settings because they are designed for classification tasks, while we need a strategy for encoder models. Tang et al. (2023) proposes an input-adaptive strategy, MuE, tailored for encoder models. MuE measures the cosine similarity between the output activations from two consecutive transformer layers to determine when to stop forward pass. If the cosine similarity becomes greater than a predefined threshold, MuE stops forwarding and subsequent layers are skipped. We test the MuE strategy to the ViT model in HAMT and evaluate both the performance and GFLOPs of the agent on the validation (unseen) set of R2R.

| Method | Performance | | | | GFLOPs(↓) |
|--------|------|--------|-------|--------|-----------|
| | TL(↓) | OSR(↑) | SR(↑) | SPL(↑) | |
| Base | 11.53 | 74.29 | 66.16 | 61.49 | 4763.24 |
| MuE | 17.37 | 62.20 | 43.93 | 36.92 | 4409.62 |

Table 1: **Performance and computational savings in HAMT with MuE.** Our adaptation of MuE leads to only marginal computational savings at the cost of significant performance degradation.

**Results.** Table 1 presents a comparison of performance and computational savings between the original HAMT and HAMT with MuE. We set the early-exit threshold to 0.998 for MuE, as the value offers the best performance-efficiency trade-off (see Appendix B for details on how we choose this threshold). We find a significant performance reduction across four metrics (up to 40%), when the HAMT agent employs MuE, while GFLOPs is only reduced by 7%. The average GFLOPs per step for the ViT in MuE applied HAMT is 406.10, compared to 607.06 in baseline HAMT. However, despite the significant per step GFLOPs savings, the total GFLOPs per trajectory increases because the MuE applied agent takes more steps to complete each trajectory.

In Figure 3, we analyze the factors contributing to the performance loss and the limited computational savings. The left figure compares trajectories from two different agents: one from the original HAMT agent and the other with MuE. Both agents navigate to the same position until $t = 2$. At $t = 3$, the original HAMT agent correctly identifies the bathroom (indicated by a green circle in the top-right figure) and successfully navigates to the front of it. But the MuE agent takes a small step forward toward it at $t = 3$. The MuE agent continues to make incorrect navigation steps until it reaches the maximum allowed steps and eventually stops. Processing fewer transformer layers can lead to an inaccurate understanding of the visual surroundings. As shown in the bottom-right figures, while the bathroom is consistently visible across multiple steps ($t \in [2, 10]$) in the panorama, the MuE agent fails to recognize it and makes sub-optimal decisions at each navigation step. In Appendix B we provide further discussion on why MuE is unsuccessful when applied directly to VLN.

## 4.2 OUR NOVEL INPUT-ADAPTIVE NAVIGATION METHOD FOR VLN

Prior work on input-adaptive inference considers models that treats each input independently. This results in existing methods inheriting the *one-size-fits-all* philosophy: a model adopts a single set

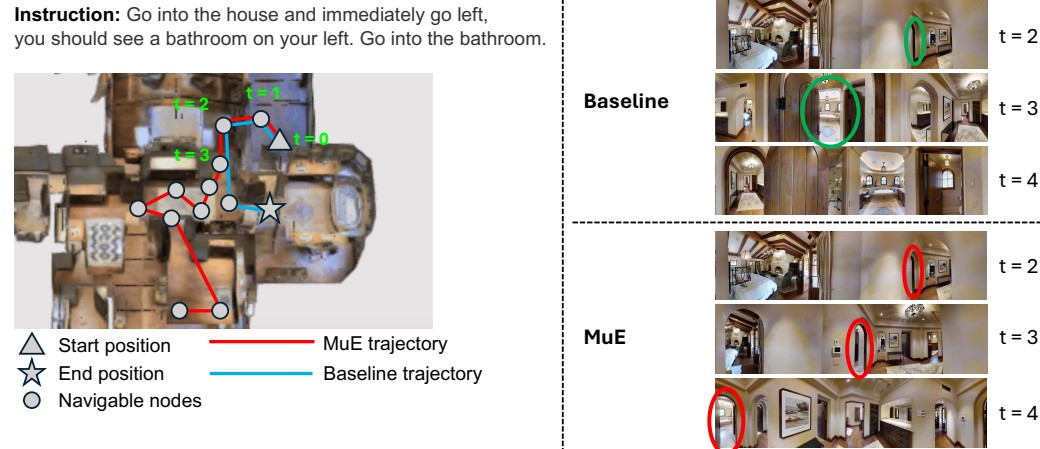

Figure 3: **Problems in employing existing input-adaptive methods in VLN.** We show that employing existing strategies lead to performance loss and an increase in computations. (Left) The increase in computations stems from inappropriate navigation actions, and (Right) such decisions come from the inaccurate understanding of the visual world, e.g., the agent confuses where to navigate.

of configurations, such as the threshold for stopping the forward pass, for all inputs. However, in dynamic settings, such as an agent navigating the physical world, the inputs are not independent to each other. They are dependent on each other both spatially and temporarily.

We introduce a novel input-adaptive inference method that harnesses this unique property—spatial and temporal dependencies in the input. We first leverage spatial locality (Sec 4.2.1): among the 36 views observed by an agent at each navigation step, we find that those close to *navigable views*—views the agent can navigate to—are important. We then propose a novel approach to assign the exit thresholds of an existing input-adaptive inference method (Sec 4.2.2) for the non-masked views to provide additional computational savings. In Sec 4.2.3, we exploit temporal locality: between panorama views observed across navigation steps, the majority of views overlap and do not require their forward passes to be run again.

### 4.2.1 HARNESSING SPATIAL LOCALITY

A panorama view is composed of 36 views, and at each navigation step, the agent computes visual embeddings for them. Our first intuition is that only *navigable views* are important for successful navigation. Intuitively, these views form the agent's decision space, and the information they contain should be sufficient for choosing the proper action. To test this hypothesis, we preserve all navigable views and mask the remaining views (setting them to zero). Suppose a panorama view contains $n$ navigable views; we mask all $36 - n$ views and keep the $n$ navigable views. This way, we can prevent the ViT model from processing these masked views, thus offering computational savings. We evaluate the effectiveness of this approach with the HAMT agent on the validation (unseen) set of R2R. We found that it resulted in an 84% gain in efficiency, but at the cost of a 33% reduction in SR.

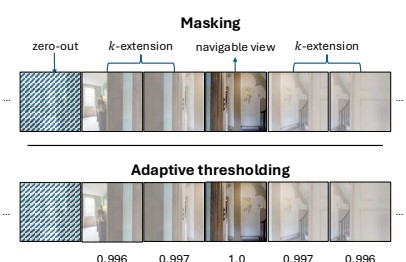

Figure 4: **Our masking and thresholding.** The top figure shows how we mask non-navigable views, and the bottom figure shows how we adaptively assign the exit thresholds of MuE.

To understand this issue, we manually analyze the failures where the agent could not reach the target after masking out non-navigable views. In Figure 4, if the agent processes only the navigable view (4th from the left), it may struggle to identify whether the pathway leads to a stairway. However, processing neighboring views increases the likelihood of correctly recognizing the pathway.

**$k$-extension.** We extend the number of views the agent processes near the navigable views by $k$. Suppose a set of $n$ navigable views in a panorama is $V$, where each navigable view $v_i$ is indexed by $\{1, 2, \dots, n\}$. The set of views in the $k$-extension $V_k^i$ for the $i$-th view is:

$$V_k^i = \{v_i^j | max(1, i - k) \leqslant j \leqslant min(i + k, 36)\},$$

where $v_i^j$ is a non-navigable view within the $k$-extension. Then the union $V_k$ of $V_k^i$s where $i \in [1, n]$, is the total views in a panorama to process, and the remaining number of masked views which we do not process is $36 - |V_k|$. With a careful calibration of $k$, we reduce the total computations by $2\times$ times while keeping the performance drop within 10%.

### 4.2.2 USING ADAPTIVE THRESHOLDS AS STOPPING CRITERIA

Now on top of our $k$-extension, we design an adaptive mechanism to early-exit the extended views and further improve model-level computational efficiency. As we describe in the previous sections, we focus on MuE, the only early-exit mechanism compatible with encoder models. All other early-exits are developed for classification tasks (or decoder models for natural language processing).

**Using the budgeted-batch inference.** The current implementation of MuE processes each test sample with its input-adaptive mechanism. However, this *per-sample, anytime* strategy is incompatible with our scenario, where the agent processes a batch of 36 views in a single panorama at once with the ViT. While each view in the batch should ideally exit at different layers, this per-sample approach forces all the views to use the same exit layer. To address this issue, we employ the concept, *budgeted-batch inference*, proposed by Huang et al. (2018). It allows each sample in a batch to use "uneven" computations, meaning that processing can stop at different layers for each sample, all within a fixed computational budget. We assign a sufficiently large computational budget so that the mechanism can handle the worst-case complexity, where none of the samples in a batch utilize early stopping.

**Our adaptive thresholding.** In Sec 4.2.1, we find that navigable views are the most important, and the importance of a view decreases as it gets farther from the navigable views in a panorama. We thus design a mechanism to apply the early-exit threshold differently based on the importance of each view. We propose a concept, *rank*: for views with a low rank, we use an aggressive early-exit with a smaller threshold, while for high-ranked views, we use a conservative (higher) threshold. Suppose we have a navigable view $v_i$ at index $i$ in a panorama and $k$ is the number of extended views near $v_i$. We define the rank $R_{i,j}$ of a non-navigable view $v_j$ relative to $v_i$ as the difference between the indices $|j - i|$. Note that we do not process when $R_{i,j} \geqslant k$, as views beyond the extension $k$ will be masked out. We still fully process the navigable views to retain performance. We then assign the exit threshold $T_{i,j}$ (the cosine similarity) for MuE as follows:

$$T_{i,j} = T_0 \cdot e^{(-A \cdot R_{i,j})}$$

where $T_0$ is the initial threshold set to 1.0, $A$ is the aggressiveness we set to $9 \times 10^{-4}$, and $R_{i,j}$ is the rank computed above. The threshold decreases as the rank increases $T_{1,j} \geqslant T_{2,j} \cdots$. In our evaluation, setting $k = 4$ typically results in sufficient computational savings with good performance.

### 4.2.3 HARNESSING TEMPORAL LOCALITY

Our final insight is that during navigation, an agent will encounter similar views multiple times, leading to *temporal redundancy*. For example, the views at navigation step $i$ will not differ significantly from those at step $i + 1$. The agent may also revisit the same surroundings due to misleading navigation or encounter similar-looking but less important surroundings, such as ceilings or walls.

To reduce this temporal redundancy, we employ locality-sensitive hashing (LSH). LSH works as a *caching* mechanism: instead of processing similar views repetitively with the ViT, it stores views and their visual representations encountered in previous navigation steps, and retrieves them at the current step. We use the hashing algorithm SimHash, which employs the random projection (Charikar, 2002; Andoni & Indyk, 2008). SimHash maps high-dimensional views ($3 \times 224 \times 224$ in our experiments) into low-dimensionality binary encodings. Given a view $v$ and the randomly initialized hyperplanes $h_i$, where $i \in \{1, \ldots, n\}$, the algorithm computes which side of the hyperplanes the view $v$ is. If $v$ is on the top side of $h_i$, SimHash assigns 1; otherwise, it is 0. Similar views are then encoded as the same binary encoding of length $n$, e.g., $010 \ldots 1$, and we use this as the key for storing pairs of views and their visual encoding. We set $n$ to 10. Like early-exiting, we refrain from hashing the navigable views and fully process them. Combined with our $k$-extension, we significantly reduce the space complexity increased by caching. We store only the subset of views processed by the ViT for each trajectory. With this mechanism, we achieve an additional 2% computational savings without utility loss. Please refer to Appendix C for more details and an analysis of the added storage overhead.

### 4.3 PUTTING ALL TOGETHER

Now we describe how the previous three mechanisms are combined to perform input-adaptive inference on a panorama view. We show the pseudo-code of our method in Algorithm 1:

---

**Algorithm 1** Our Input-adaptive Navigation at Each Step

---

**Input:** a panorama $P$, a set of navigable views $V$, a ViT $f_\theta$, a hash table $h$, and the number of views to extend $k$
**Output:** a set of visual representations $E$ for views in $P$

1: $E \leftarrow \varnothing$
    # loop over the views in $P$
2: **for** $i = 1, 2, \ldots, 36$ **do**
3:     $v_i \leftarrow P[i]$
4:     **if** $v_i$ in $V$ **then**
5:         $e_i \leftarrow f_\theta(v_i)$
6:         $E \leftarrow E + e_i$
7:     **else if** $i$ in $k/2$ proximity of any views in $V$ **then**
8:         $e_i \leftarrow h(v_i)$
9:         **if** $e_i$ does not exist **then**
10:             $j \leftarrow$ the index of the closest navigable view
11:             $T_i \leftarrow \texttt{ComputeThreshold}(R_{i,j})$
12:             $e_i \leftarrow \texttt{RunMuEInference}(v_i, T_i)$
13:             $h \leftarrow \texttt{AddToHashTable}(h, v_i, e_i)$
14:         **end if**
15:         $E \leftarrow E + e_i$
16:     **else**
17:         $E \leftarrow E + \vec{0}$
18:     **end if**
19: **end for**
20: **return** $E$

---

**(line 1-2) Initialize.** It takes a panorama view $P$ and returns the visual representations of the 36 views composing $P$. We initialize the output set $E$ as $\varnothing$ and start iterating over each of the 36 views.

**(line 4-6) Compute the representation of a navigable view.** Suppose the view currently chosen $v_i$ is a navigable view. We fully compute its visual representation $e_i$ and add it to the set $E$.

**(line 7-15) Retrieve (or compute) the representation of the extended views.** In Sec 4.2.1, to improve the understanding of the visual world, we develop the $k$-extension. We process $k/2$ views on both sides (left and right) of a navigable view. If we have the representation of $v_i$ in the hash table $h$, then we retrieve $e_i$ and add it to $E$. Otherwise, we compute $e_i$. Note that the hash table $h$ is initialized at the very first step of the navigation. To compute $e_i$, we first determine $v_i$'s rank $R_{i,j}$ and decide the exit threshold $T_i$. We run the inference with ViT, adapted for MuE, using $T_i$ and store the output $e_i$ into $h$ and $E$.

**(line 17) Skipping the masked view.** If $v_i$ is neither a navigable view nor in its $k$-extension, we skip processing the view by storing a zero-vector and move on to the next view $v_{i+1}$.

## 5 EMPIRICAL EVALUATION

### 5.1 EFFECTIVENESS OF OUR INFERENCE METHOD

We first compare the computational savings our method offers in two agents across six popular benchmarking tasks. As we describe in Sec 3, we measure four performance metrics along with GFLOPs to quantify the computation an agent requires to finish navigation. We compare with two baselines: Base, without any input-adaptive method and MuE, where we adapt an existing method for each agent to provide the optimal performance-efficiency trade-off. In our method, we present four variations: one with $k$-

| Agent | Method | Performance | | | | GFLOPs($\downarrow$) |
|-------|--------|------|------|------|------|------|
| | | TL($\downarrow$) | OSR($\uparrow$) | SR($\uparrow$) | SPL($\uparrow$) | |
| HAMT | Base | 11.53 | 74.29 | 66.16 | 61.49 | 4763.24 |
| | MuE | 17.37 | 62.20 | 43.93 | 36.92 | 4409.62 |
| | Ours ($k$-extension) | 12.52 | 71.86 | 61.30 | 55.79 | 2408.99 |
| | Ours ($k$-extension+LSH) | 12.52 | 71.90 | 61.17 | 55.63 | 2013.48 |
| | Ours ($k$-extension+thresholds) | 12.89 | 71.95 | 60.41 | 54.57 | 2294.23 |
| | Ours (All) | 12.87 | 71.95 | 60.41 | 54.5 | 1917.61 |
| DUET | Base | 13.94 | 81.10 | 71.73 | 60.57 | 4998.00 |
| | MuE | 16.88 | 74.84 | 62.24 | 49.99 | 4424.21 |
| | Ours ($k$-extension) | 14.03 | 75.22 | 65.69 | 54.06 | 2413.11 |
| | Ours ($k$-extension+LSH) | 14.01 | 75.18 | 65.73 | 54.17 | 2171.84 |
| | Ours ($k$-extension+thresholds) | 14.21 | 73.82 | 63.39 | 52.21 | 2254.82 |
| | Ours (All) | 14.21 | 73.86 | 63.47 | 52.35 | 2026.30 |

Table 2: **Effectiveness of our input-adaptive inference method.** We show our results on R2R for both the HAMT and DUET agents. Each cell contains the averaged metric over the trajectories in the validation (unseen) set. Our method achieves $\sim 60\%$ computational savings with a marginal performance loss of $\sim 10\%$ (in SR).

extension ($k = 4$), and two others (+LSH or +thresholds) that allow us to quantify the effectiveness of the other mechanisms we design on top of the $k$-extension, and the final one where we apply All.

**Results.** Table 2 summarizes our results in R2R. Due to the page limit, we show the full results for other tasks in Appendix D and more combinations of mechanisms in Appendix E. In R2R, our method which combines $k$-extension, LSH, and adaptive thresholding saves ~60% computation while maintaining an SR loss between 8.6–11.5%. We set the upper-limit for performance loss near 10%, consistent with prior work on input-adaptive inference methods (Huang et al., 2018; Liu et al., 2020; Kaya et al., 2019; Xin et al., 2020; Tang et al., 2023). The naive adaptations of MuE on both agents only provide 7.4–11.5% computational savings and experience a significant performance drop of 13.2–33.6% in SR, as expected from our initial investigation in Sec 4.1. Our $k$-extension alone provides a 49.4–51.7% reduction in GFLOPs with only a 7.3–8.5% drop in SR. If we apply the adaptive thresholding (+thresholds), we achieve an additional 2.4–3.2% computational savings, with a marginal performance loss of ~1%. Separately, incorporating the LSH into the $k$-extension results in an additional computational savings up to 1.9%, with no performance loss (even the SR increases).

## 5.2 SENSITIVITY TO OUR METHOD'S CONFIGURATIONS

Next we evaluate how sensitive the computational savings provided by our method to its configurations. If a method is too sensitive to the configuration choices, it increases the *hidden cost* of employing our approach in practice. We note that no prior work so far has discussed the cost of employing input-adaptive inference methods. Our method has three key configurations that can impact its effectiveness: the number of extended views ($k$), the adaptive thresholds set based on the extension, and the similarity measure used in our LSH mechanism. Here we show our results in R2R.

| $k$ | Performance | | | | GFLOPs($\downarrow$) |
|---|---|---|---|---|---|
| | **TL($\downarrow$)** | **OSR($\uparrow$)** | **SR($\uparrow$)** | **SPL($\uparrow$)** | |
| - | 11.53 | 74.29 | 66.16 | 61.49 | 4763.24 |
| 1 | 15.38 | 70.20 | 54.32 | 46.96 | 1250.65 |
| 2 | 13.67 | 70.84 | 58.19 | 51.99 | 1554.82 |
| 3 | 12.94 | 71.60 | 60.20 | 54.60 | 1793.76 |
| 4 | 12.52 | 71.90 | 61.17 | 55.63 | 2013.48 |
| 5 | 12.19 | 71.99 | 62.32 | 57.08 | 2216.34 |
| 6 | 11.89 | 71.99 | 62.84 | 57.94 | 2414.46 |

Table 3: **Performance and computational savings across different $k$ values.** We evaluate with the HAMT agent in R2R. $k$ is chosen in [1, 6].

**Number of extended views ($k$).** Table 3 summarizes the variations in performance and GFLOPs for different numbers of extended views. We vary $k$ in [1, 6]. As $k$ decreases, the agent only needs to process a few views in each panorama, which results in computational savings of 49.3–73.7% and a performance drop of 5–18%. It is surprising that with $k = 1$, we can save 74% of computations while only sacrificing 18% in performance (SR). We choose $k$ such that an agent processes approximately half of the total 36 views in each panorama on average. In R2R, we find that each panorama has an average of 4 navigable views. Extending each of them to 4 neighboring views then results in ~18 views per panorama. Given that this strategy provides 50% computational savings across all benchmarks, even when the average number of navigable views per panorama is not considered for setting $k$, we believe the strategy is transferable to new settings.

| $A$ | Thresholds $T$ | | | | Performance | | | | GFLOPs |
|---|---|---|---|---|---|---|---|---|---|
| | $R_{1,j}$ | $R_{2,j}$ | $R_{3,j}$ | $R_{4,j}$ | **TL($\downarrow$)** | **OSR($\uparrow$)** | **SR($\uparrow$)** | **SPL($\uparrow$)** | |
| 0 | 1.0 | 1.0 | 1.0 | 1.0 | 12.52 | 71.90 | 61.17 | 55.63 | 2013.48 |
| 0.007 | 1.0 | 1.0 | 1.0 | 0.997 | 12.57 | 71.60 | 60.96 | 55.32 | 1973.23 |
| 0.009 | 1.0 | 1.0 | 0.997 | 0.996 | 12.87 | 71.95 | 60.41 | 54.5 | 1917.61 |
| 0.015 | 1.0 | 0.997 | 0.996 | 0.993 | 13.44 | 70.67 | 57.98 | 52.09 | 1848.89 |
| 0.022 | 0.997 | 0.996 | 0.993 | 0.990 | 14.61 | 70.29 | 55.60 | 48.56 | 1768.85 |

Table 4: **Performance and computational savings across different early-exit thresholds.** We set the aggressiveness $A$ within [0.0, 0.022]. Note that we round the threshold to 3 decimal places and set any thresholds greater than 0.998 to 1.0 as ViTs with these thresholds will use full computations.

**Early-exit thresholds.** We also analyze the impact of different early-exit thresholds $T$. To this end, we vary the aggressiveness factor $A$ from 0.0 to 0.0022. Our method decreases the threshold as a view becomes farther from a navigable view. Table 4 shows that as the early-exit becomes more aggressive (progressing from the top to the bottom rows), performance decreases while computational efficiency increases. We find that using the aggressiveness over 0.009 leads to a significant performance drop in SR over 10%. We thus set $A$ to 0.099.

**Using different similarity metrics.** In Sec 4.2.3, our primary metric for computing similarity between views is cosine similarity based on raw RGB pixel values. We explore whether employing different similarity metrics, commonly used in computer vision studies, can further enhance the effectiveness of our method. To evaluate this, we test four additional metrics: visual features extracted from ViT's first-layer activations, SSIM (Wang et al., 2004), FSIM (Zhang et al., 2011), and LPIPS (Zhang et al., 2018). We also test other metrics, e.g., SURF (Bay et al., 2006) or SIFT (Lowe, 2004), but they fail to match even visually similar views in consecutive navigation steps (see Appendix G for more details).

Table 5 shows our results. Across the board, we observe only a marginal difference between the similarity metrics. We see a performance increase of 0.16–0.22% at the cost of a 2.6% increase in computation. The largest increase in computation comes from running forward to obtain the intermediate activation from ViT (the second row from the top). The results indicate that our method is not dependent on the choice of similarity metrics, studied so far in prior work. We also manually analyze the views identified as similar by these metrics, and most of the views selected are either identical or had tiny variations, e.g., plain walls with lighting differences.

| Similarity Metrics | Performance | | | | GFLOPs |
| --- | --- | --- | --- | --- | --- |
| | TL | OSR | SR | SPL | |
| **RGB (Ours)** | 12.87 | 71.95 | 60.41 | 54.50 | 1917.61 |
| **ViT (1st layer activation)** | 12.89 | 71.99 | 60.41 | 54.59 | 1966.95 |
| **SSIM** (Wang et al., 2004) | 12.87 | 71.95 | 60.41 | 54.57 | 1934.48 |
| **FSIM** (Zhang et al., 2011) | 12.88 | 71.95 | 60.45 | 54.58 | 1937.73 |
| **LPIPS** (Zhang et al., 2018) | 12.87 | 71.95 | 60.49 | 54.62 | 1925.15 |

Table 5: **Impact of employing different similarity metrics in LSH.** We experiment with the HAMT model in R2R.

### 5.3 ROBUSTNESS OF OUR METHOD UNDER NATURAL CORRUPTIONS

Recent work has explored the robustness of VLN agents to various natural corruptions in their environments (Chattopadhyay et al., 2021). We thus evaluate how robust the computational savings our approach offers are to various visual corruptions that can happen in practice. We select five common corruptions that an agent may encounter in real-world navigation scenarios: Spatter, Defocus Blur, Speckle Noise, Low Lighting, and Motion Blur. Figure 5 shows how each corruption visually affects a scene. We apply each corruption to the entire validation (unseen) set of R2R, using the corruption framework by Chattopadhyay et al. (2021). We set the severity to 3 out of 5 because setting the severity level above 3 causes excessive distortion to the views, which does not reflect the realistic corruptions an agent would encounter.

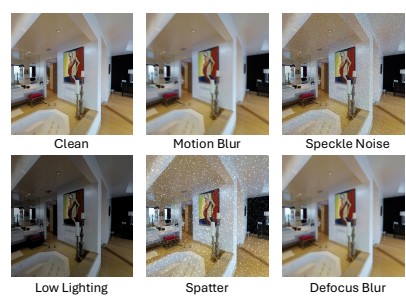

Figure 5: **Example views illustrating the five visual corruptions we consider.**

| Agent | Corruption | Performance | | | | GFLOPs(↓) |
| --- | --- | --- | --- | --- | --- | --- |
| | | TL(↓) | OSR(↑) | SR(↑) | SPL(↑) | |
| HAMT | None | 11.53 | 74.29 | 66.16 | 61.49 | 4763.24 |
| | Spatter | 13.30 | 69.82 | 58.71 | 52.91 | 5227.36 |
| | Defocus Blur | 13.87 | 66.50 | 55.21 | 49.32 | 5383.35 |
| | Speckle Noise | 13.60 | 62.88 | 51.68 | 46.02 | 5345.07 |
| | Low Lighting | 12.15 | 71.31 | 62.58 | 57.23 | 4903.06 |
| | Motion Blur | 12.41 | 68.20 | 59.13 | 54.01 | 4996.64 |
| Ours | None | 12.87 | 71.95 | 60.41 | 54.50 | 1917.61 |
| | Spatter | 16.09 | 67.01 | 49.04 | 41.53 | 2201.19 |
| | Defocus Blur | 16.22 | 63.69 | 49.21 | 41.73 | 2082.57 |
| | Speckle Noise | 18.11 | 61.43 | 40.87 | 33.60 | 2342.67 |
| | Low Lighting | 15.27 | 69.90 | 52.58 | 45.33 | 1516.50 |
| | Motion Blur | 14.47 | 65.47 | 52.96 | 46.52 | 1986.50 |

Table 6: **Robustness evaluation of vanilla HAMT and efficient HAMT under visual corruptions** We evaluate both the Vanilla HAMT and our Efficient HAMT on the R2R dataset under clean conditions and five different types of visual corruption: spatter, defocus blur, speckle noise, low lighting, and motion blur.

**Results.** Table 6 summarizes our findings from evaluating the HAMT agent on the R2R benchmark. We first observe that applying our method to a VLN agent reduces its performance and computational savings compared to the original agent. Across the five corruptions, the HAMT agent shows 5.4–21.1% reductions in performance, while our agent undergoes 12.3–31.3% reductions. GFLOPs increase by 2.9–13.0% in HAMT, while we show an increase of 3.6–20.9%. Per corruption, we find that both agents are most resilient to Low Lighting and least robust to Speckle Noise. This aligns with the findings of prior work (Chattopadhyay et al., 2021). In our evaluation, all the agents use visual encoders pre-trained on ImageNet-

1K, meaning our agents inherit the susceptibility of these ImageNet encoders to visual corruptions. This finding highlights the importance of studies on enhancing the robustness of visual encoders to natural corruptions (Hendrycks & Dietterich, 2019; Guo et al., 2023; Zhu et al., 2023), which could improve the robustness across various VLN agents.

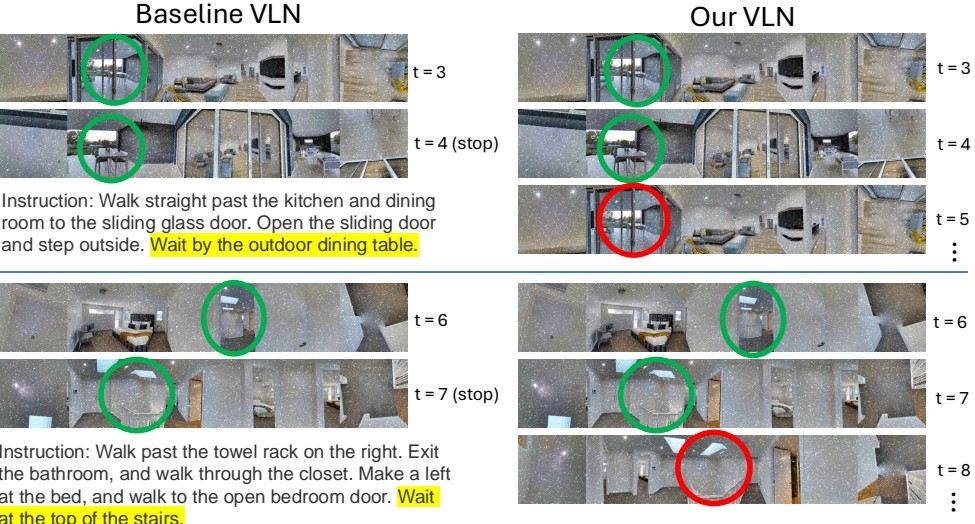

Figure 6: **Comparison of baseline and our agent trajectories under Spatter corruption.** We demonstrate that our agent fails to stop at the target location, resulting in incorrect navigation (Right), whereas the baseline agent successfully stops as instructed (Left).

We conduct a manual analysis of the trajectories from the same task for the baseline and our agents in HAMT. We run this analysis in R2R, where we observe that our HAMT agent suffers from lower robustness compared to the baseline agent when exposed to visual corruptions. Interestingly, we find that while our agent experiences a significant drop in SR, the degradation in OSR is notably smaller. Figure 6 illustrates these examples. In the top figure, our agent fails to stop at the target location, while the baseline agent successfully stops as intended. The agent is supposed to stop in front of the outdoor dining table, but our agent reaches the target point and turns around instead of stopping. It then continues turning until it reaches the maximum allowed steps. Similarly, in the example shown at the bottom, both agents are required to stop at the top of the stairs, but our agent passes the stairs and heads into the bedroom. This finding indicates that our agent can navigate to the target destination, but it struggles to stop correctly at the location in the presence of visual corruptions. Navigating along the correct trajectory and stopping at the precise location may require different capabilities: the former relies on pathway identification, while the latter depends on object recognition. Thus, we hypothesize that our approach does not degrade an agent's ability to navigate, but negatively impacts its ability to recognize where they are. Because the robustness of object recognition under various visual corruptions is a separate, active area of research, we decide to leave this for future work.

## 6  CONCLUSION

We propose an input-adaptive inference method to mitigate overthinking in vision-and-language navigation (VLN) and achieve computational efficiency. Unlike the overthinking problem in conventional domains, such as object recognition or natural language comprehension, addressing overthinking in VLN presents three unique challenges: (1) How can we leverage spatial locality in views observed by an agent at a navigation step? (2) How can we reduce temporal redundancy across the agent's navigation steps? (3) How can we use the mechanisms designed to address the two challenges to adaptively set early-exit thresholds of an existing method? We present three novel techniques to address them individually. In our evaluation, we demonstrate a $2\times$ reduction in computations while preserving performance across six VLN benchmarks. Moreover, we assess the robustness of our approach under various visual corruptions that may occur in practice, and identify challenges to address for future work. We hope this work inspires future research on developing efficient (and robust) VLN algorithms and promote their widespread adoption in real-world settings.

**Reproducability Statement.** To ensure our work is reproducible, we provide comprehensive descriptions of the dataset, models, hyper-parameters and our input-adaptive inference method both in the main text and in the Appendix. Specifically, Sec 3, Sec 4.2, Sec 5.2, and Appendix A offer detailed discussions on these topics. Our proposed input-adaptive inference algorithm is presented in Algorithm 1. We believe these thorough implementation details will enable others to successfully replicate our work. Additionally, we plan to release the source code to further support the reproducibility.

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

## A  EXPERIMENT SETUP IN DETAIL

We describe the experimental setup used to evaluate our input-adaptive inference mechanism in detail. We implemented our strategy on top of the codebases provided by the authors of the HAMT (Chen et al., 2021) and DUET (Chen et al., 2022). During inference, instead of using cached image features, we integrate a ViT-B/16 (Dosovitskiy et al., 2021) model to process the images directly.

**Hardware and software.** We run our experiments on a machine equiped with an Intel Xeon processor with 48 cores, 64GB of DRAM, and 8 NVIDIA A40 GPUs, with all inference tasks performed on a single GPU using a batch size of 1. Following the original HAMT study, we use Python v3.8.5 and PyTorch v1.7.1, along with CUDA v10.1. For GFLOPs calculations, we use the Python library thop.

**Datasets.** We describe the benchmarking datasets we use in detail:

- **R2R** (Anderson et al., 2018) is based on Matterport3D (Chang et al., 2017), containing 10,567 panorama views taken from 90 photo-realistic houses. The dataset includes 7,189 shortest-path trajectories, and each of them is associated with 3 natural language instructions. The training, validation (seen), validation (unseen), and test (unseen) sets include 61, 56, 11, and 18 houses, respectively. The validation (seen) set consists of houses in the training set, typically used to check the generalization status of a model during training, while the sets marked as 'unseen' are the houses not in the training set.
- **R2R-Back** (Chen et al., 2021) requires the agent to return to its starting point after reaching the destination. To complete the task, the agent must remember its navigation history. A return command is appended to each R2R instruction, and the reversed path is provided as guidance for the return trip.
- **R2R-Last** (Chen et al., 2021) uses only the last sentence from the original R2R instructions to describe the destination.
- **REVERIE** (Qi et al., 2020) provides high-level instructions, closer to those given by humans, replacing the step-by-step instructions of R2R. Instead of navigating to a target location, the agent is required to identify and localize the target object upon arrival, making the task more complex and realistic. The dataset includes 4,140 target objects, which are categorized into 489 distinct groups.
- **CVDN** (Thomason et al., 2020) requires the agent to navigate based on long, potentially unclear instructions. The agent interacts with a navigator through question and answer dialog to clarify and complete the task. In total, it has 2,050 human-human navigation dialogues, consisting of over 7,000 navigation trajectories accompanied by question-answer interactions, covering 83 matterport3D houses.
- **SOON** (Zhu et al., 2021) is similar to REVERIE but contains longer and more detailed instructions. The average length of these instructions is 47 words, with path lengths varying from 2 to 21 steps. It requires the agent to navigate by understanding the relationship between objects in the environment to accurately locate the target object.

## B  OPTIMAL HYPERPARAMETER CHOICE FOR ADAPTING MUE TO OUR WORK

To best evaluate MuE on VLN tasks, we perform a hyperparameter sweep over the threshold used for early-exiting. Figure 7 presents the performance (in SR) and GFLOPs across different early exit thresholds applied to the MuE version of ViT used in the HAMT agent, tested on the R2R dataset. The lowest threshold we report is 0.99, as lower thresholds caused a dramatic drop in performance (more than 50%). As the threshold increases, the success rate of the MuE agent increases substantially but at the cost of computational savings. Even for thresholds close to 1, meaning that the ViT is using a majority of its layers for each input, we still see a large performance drop compared to the baseline agent. As we discuss in Sec 4.2, this is likely because MuE statically applies early-exits, causing it to under-process important components of the panorama such as navigable views.

**Why does MuE underprocess important views?** The intuition behind MuE (Tang et al., 2023) is that the activations of Transformer-based vision models *saturate*, where their similarity between layers peaks early on, and is maintained at future stages of computation, suggesting a lack of new/useful information. MuE then exploits this property to skip the later layers without a significant loss in performance. So, for MuE to be successful, the similarity of activations must sufficiently saturate and

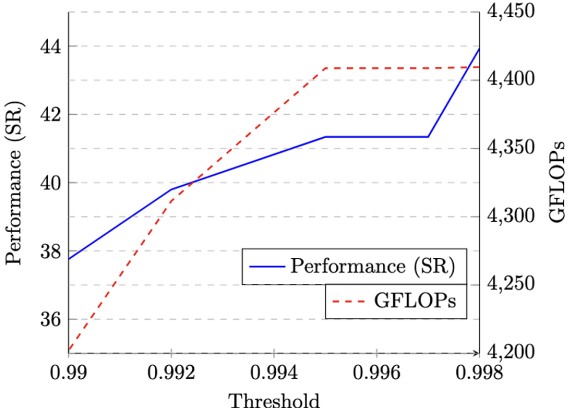 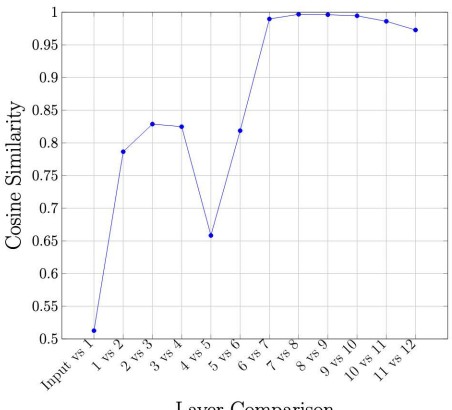

Figure 7: **Comparison of performance (in SR) and GFLOPs in MuE across different thresholds.**

Figure 8: **Cosine similarity between adjacent layers of ViT used in HAMT.**

not decrease at later layers. However, as shown in Figure 8, the necessary saturation pattern is not observed in the VLN setting. The cosine similarity peaks between layers 7 and 8 but then decreases for all future layers. This explains the significant performance drop when MuE is directly applied to VLN agents, as it consistently early-exits despite saturation not being achieved.

## C  OUR LSH ALGORITHM IN DETAIL

A core mechanism we introduce in Sec 4.2.3 is our SimHash algorithm, used to avoid reprocessing previously seen or near-identical images. Algorithm 2 covers our implementation in detail.

**(line 1-9) Hashing RGB vectors.** Given an image, we first hash the raw RGB vector into a short binary encoding using random projection Charikar (2002); Andoni & Indyk (2008). The algorithm calculates the dot product between the image vector and each hyperplane. If the dot product is positive, it assigns a binary value of 1, otherwise it assigns 0. These binary values are sequentially appended to form a complete binary hash key. The length of the hash key is determined by the number of hyperplanes used in the projection.

**(line 10-14) Adding embeddings to the hash table.** This function is used to insert processed images and their corresponding embeddings into the hash table for future use.

**(line 15-32) Retrieving a similar embedding.** This function takes an image we have not yet processed and tries to find a suitable embedding candidate. We first obtain all embeddings with images similar to the current image by hashing it into its binary encoding and accessing the corresponding bucket in the hash table. We then loop through all images associated with the similar embeddings and find the one yielding the highest similarity score (in our main experiments, the score is computed using cosine similarity). If this score exceeds our threshold hyperparameter, we return the associated embedding; otherwise, we return nothing.

**Running the algorithm.** We employ the above three functions to run SimHash on an arbitrary panorama. For each extended navigable view (other views are omitted and explained in Algorithm 1), we attempt to use a high-similarity embedding from the hash table. If it exists, we reuse this embedding for the current view and continue to the next. If not, we need to process the view using the ViT adapted for MuE, and then add the image and its embedding to the hash table. After processing the entire panorama, we return the set of final embeddings to be used for agent navigation.

**Storage overhead analysis.** Here, we consider the storage overhead necessary to deploy our hashing algorithm on VLN agents. Our LSH technique stores pairs of images and embeddings. In the benchmarks we consider, these images are of size 3x224x224. The embedding size depends on the model, which for HAMT and DUET is 197x768 (the number of ViT patches times the model's hidden dimension). These are stored in full-precision floating-point format (32 bits per value), resulting in $(3 \times 224 \times 224 + 197 \times 768) \times 32$ bits of storage per cached pair, approximately 1.2 MB. In our experiments, the longest navigation route was roughly 12 steps (from R2R-Back), and if we assume

---

**Algorithm 2** SimHash Algorithm

---

**Input:** a current view $v_i$
**Output:** a binary hash $key$
 1: **function** HASH($v_i$)
 2:     $key \leftarrow \varnothing$
 3:     **for** each $hp$ **in** Hyperplanes **do**
 4:         $sign \leftarrow$ DotProduct($hp, v_i$)
 5:         $hash\_val \leftarrow (sign > 0)$                                   ▷ converts to binary
 6:         $key \leftarrow key + hash\_val$
 7:     **end for**
 8:     **return** $key$
 9: **end function**
**Input:** a hash table $h$, a current view $v_i$, an embedding $e_i$
**Output:** a hash table $h$
10: **function** ADDTOHASHTABLE($h, v_i, e_i$)
11:     $key \leftarrow$ Hash($v_i$)
12:     $h \leftarrow$ InsertToHashTable($key, v_i, e_i$)
13:     **return** $h$
14: **end function**
**Input:** a hash table $h$, a current view $v_i$
**Output:** an embedding $e_i$
15: **function** FINDSIMILAR($h, v_i$)
16:     $s_{max} \leftarrow -1$
17:     $key \leftarrow$ Hash($v_i$)
18:     $bucket \leftarrow h.get(key)$
19:     **for** each $(v_{candidate}, e_{candidate})$ **in** bucket **do**
20:         $s \leftarrow$ **CosineSimilarity**($v_i, v_{candidate}$)
21:         **if** $s > s_{max}$ **then**
22:             $s_{max} \leftarrow s$
23:             $e_{best} \leftarrow e_{candidate}$
24:         **end if**
25:     **end for**
26:     **if** $s_{max} > threshold$ **then**
27:         $e_i \leftarrow e_{best}$
28:     **else**
29:         $e_i \leftarrow \varnothing$
30:     **end if**
31:     **return** $e_i$
32: **end function**

---

all 36 images per panorama are cached, we obtain a worst-case overhead of 522.7 MB. In practice, however, we find that most tasks are 5–7 steps, and we cache at most 14 images per step, producing a more typical overhead of 84.7–118.6 MB. Considering that modern VLN agents Chen et al. (2021; 2022) are orders of magnitude larger, this is not a limiting factor to practical deployment.

## D  FULL EVALUATION RESULTS

Table 7 complements our main evaluation in Sec 5.1 with additional benchmarks: R2R-Back (Chen et al., 2021), REVERIE (Qi et al., 2020), R2R-Last (Chen et al., 2021), CVDN (Thomason et al., 2020), and SOON (Zhu et al., 2021). For CVDN, we report the additional evaluation metric Goal Progress (GP), which assigns a higher score as the agent moves closer to the goal, indicating better performance (Chen et al., 2021). For REVERIE and SOON, in addition to image features, object features are required during navigation. We were unable to find the original implementation for object feature extraction, so for these benchmarks we use cached object features and apply our strategy only to image feature extraction. To accommodate this in the performance calculations, we report the GFLOPs necessary for image feature processing and treat the computational cost of object feature extraction as a constant (the $+C$ in Table 7). Note that this prevents us from being able to report

| Agent | Task | Method | Performance | | | | | GFLOPs |
|-------|------|--------|-----|-----|-----|-----|-----|--------|
| | | | TL | OSR | SR | SPL | GP | |
| HAMT | R2R-Back | Base | 20.56 | - | 55.43 | 52.34 | - | 8181.55 |
| | | Ours (All) | 20.53 | - | 49.21 | 46.47 | - | 3331.80 |
| | REVERIE | Base | 14.07 | 35.73 | 31.81 | 29.17 | - | 5434.71+$C$ |
| | | Ours (All) | 13.70 | 26.75 | 24.96 | 23.13 | - | 2735.90+$C$ |
| | R2R-Last | Base | 12.28 | 54.24 | 47.85 | 42.27 | - | 4982.68 |
| | | Ours (All) | 12.36 | 49.72 | 41.93 | 36.97 | - | 2589.44 |
| | CVDN | Base | - | - | - | - | 4.88 | 11022.03 |
| | | Ours (All) | - | - | - | - | 4.45 | 4773.34 |
| DUET | REVERIE | Base | 22.49 | 51.46 | 47.09 | 33.54 | - | 6185.15+$C$ |
| | | Ours (All) | 21.59 | 46.44 | 41.32 | 28.90 | - | 3350.31+$C$ |
| | SOON | Base | 35.87 | 50.38 | 36.19 | 22.67 | - | 9997.81+$C$ |
| | | Ours (All) | 42.36 | 54.22 | 36.43 | 20.37 | - | 4533.83+$C$ |

Table 7: Comparison of the performance and efficiency of the baseline agents versus our improved-efficiency agents across multiple benchmarks. Here, we denote the cost of object feature extraction as $C$.

percentage-wise changes in total performance, so we consider the raw reduction in GFLOPs in these cases.

The upper section of the table compares the performance and efficiency of the baseline HAMT agent against our efficient HAMT agent. For R2R-Back, our strategy achieves a 60% reduction in computation with an 11% decrease in SR. For REVERIE, our efficient VLN model reduces computation by 2698.81 GFLOPs, with a 20% drop in SR. For R2R-Last, our method reduces computation by 48%, with a 12% reduction in SR. Finally, for the CVDN evaluation, our efficient model reduces computation by 57%, with only a 9% decrease in GP. The lower section of the table presents a comparison of the performance and efficiencies of the DUET agents. For REVERIE, our strategy saved 2834.84 GFLOPs with a 12% decrease in SR. For SOON, we observed a marginal increase in SR accompanied by a 10% drop in SPL, while saving 5463.98 GFLOPs. Despite the more significant performance drop in the REVERIE task using the HAMT agent, these results demonstrate that our efficiency strategies are applicable across different benchmarks, achieving substantial computational savings while maintaining an acceptable trade-off in performance.

**Robustness to navigation length.** It is possible that the errors introduced by our method *propagate*, resulting in worse agent navigation for longer trajectories. We study if this is the case by considering the *navigation error* (NE)—the distance of an agent's final position to the target position (in meters)—on benchmarks with varying path lengths. We deploy all of our proposed methods (simultaneously) on the HAMT agent and report the changes in NE and GFLOPs compared to the baseline in Table 8.

| Agent | Task | Average Path Length | $\Delta$NE($\downarrow$) | $\Delta$GFLOPs($\downarrow$) |
|-------|------|---------------------|--------|----------|
| HAMT | R2R | 6.0 | +0.53 | -2845.63 |
| | R2R-Last | 6.0 | +0.45 | -2393.24 |
| | R2R-Back | 12.0 | +0.54 | -5463.98 |
| DUET | R2R | 6.0 | +0.68 | -2971.70 |
| | SOON | 9.6 | -0.44 | -5463.98 |

Table 8: Performance of our efficient HAMT agent on benchmarks with different path lengths. $\Delta$NE and $\Delta$GFLOPs are the changes in navigation error (NE) and GFLOPs compared to the baseline agent. The path length is the minimum number of navigation actions needed to reach the target destination.

We find our method is largely robust to longer path lengths. The NE does not increase for longer trajectories, and we even see a decrease for the SOON benchmark, which has an average path length 3.6 more steps than R2R. The results also show that our efficient VLN agent sees roughly proportional computational savings for longer paths. For example, the average path length in R2R-Back is double R2R, and we achieve a 1.92x larger reduction in GFLOPs for the HAMT agent.

| Method | TL($\downarrow$) | OSR($\uparrow$) | SR($\uparrow$) | SPL($\uparrow$) | GFLOPs($\downarrow$) |
|---|---|---|---|---|---|
| None (Base) | 11.53 | 74.29 | 66.16 | 61.49 | 4763.24 |
| $k$-extension | 12.52 | 71.86 | 61.30 | 55.79 | 2,408.99 |
| thresholds | 12.33 | 72.46 | 62.62 | 57.39 | 3,867.46 |
| LSH | 11.53 | 74.20 | 66.11 | 61.47 | 3,894.76 |
| $k$-extension+LSH | 12.52 | 71.90 | 61.17 | 55.63 | 2,013.48 |
| $k$-extension+thresholds | 12.89 | 71.95 | 60.41 | 54.57 | 2,294.23 |
| thresholds+LSH | 12.33 | 72.41 | 62.49 | 57.33 | 3,190.66 |
| All | 12.87 | 71.95 | 60.41 | 54.50 | 1,917.61 |

Table 10: Performance of all combinations of our speed-up techniques ($k$-extensions, early-exiting, and LSH) with the HAMT agent on the R2R benchmark.

**Runtime comparison.** To validate that our approach improves efficiency in the real world, we report the wall-time comparison between our efficient VLN model and the baseline VLN for both HAMT and DUET agents, tested on the R2R validation unseen split, in Table 9. Evidently, our efficient strategy applied to the VLN agents results in significant runtime savings, with an approximate 40% reduction. It is important to note that the disparity between the 60% GFLOPs savings and the 40% runtime reduction can be attributed to various hardware and software related factors.

| Task | Agent | Method | Wall-time (s) |
|---|---|---|---|
| **R2R** | HAMT | Base | 200811 |
| | | Ours | 119514 |
| | DUET | Base | 268962 |
| | | Ours | 170464 |

Table 9: Wall-time comparison between the baseline agent and our efficient agent on the R2R task.

# E  PER-MECHANISM ANALYSIS

In most experiments, we treat our proposed mechanisms as a single unit by applying all three simultaneously. While this is the most flexible and offers the best trade-off between performance and efficiency, analyzing each mechanism independently can provide valuable insights into its effectiveness and robustness. Here, we present results on a per-mechanism basis.

**Effectiveness.** In Sec 5.1, we apply our $k$-extension technique and then add adaptive thresholding early-exiting (denoted thresholds in Table 2) and locality-sensitive hashing (LSH) as we found those combinations of techniques offer the most computational savings. Here, we study all combinations of three efficiency mechanisms. To use early-exiting and LSH without $k$-extension, we treat every non-navigable view as one that can be early-exited or hashed. Navigable views are still fully processed. We report results for the HAMT agent on the R2R benchmark in Table 10.

The results show that between individual techniques, $k$-extension offers the best computational savings with a 49% reduction compared to the baseline agent. Early-exiting and LSH only reduce GFLOPs by $\sim$18% because early-exiting still requires processing every view, and LSH reuses only a minority of cached image embeddings. We find that LSH provides better performance than the other two individual mechanisms, with an SR only 0.05 lower than the baseline. This is likely because the cached embeddings reused by LSH are near-identical, having a negligible impact on performance when interchanged. However, it is far less efficient than when combined with our other techniques.

The combination we do not present in Table 2, early-exiting and LSH (**thresholds+LSH**), provides slightly better performance than combinations using $k$-extension but at the cost of 39–66% more GFLOPs. Like the individual mechanisms, this suggests that retaining and partially processing/reusing the non-navigable views mitigates performance drop but is not nearly as efficient as $k$-extension. Overall, we find that all combinations of our techniques fare well, offering different trade-offs between performance and efficiency.

**Robustness to natural corruptions.** Now, we complement Sec 5.3 and study the robustness of each of our proposed mechanisms to visual corruption. We select the Low Lighting and Motion Blur corruptions based on their varying impact on performance and being more likely to occur in

| Corruption | Method | TL($\downarrow$) | OSR($\uparrow$) | SR($\uparrow$) | SPL($\uparrow$) | GFLOPs($\downarrow$) |
|---|---|---|---|---|---|---|
| | None (Base) | 12.15 | 71.31 | 62.58 | 57.23 | 4903.06 |
| Low Lighting | $k$-extension | 13.86 | 71.14 | 57.34 | 50.78 | 2571.06 |
| | thresholds | 13.63 | 70.29 | 58.79 | 52.16 | 4099.21 |
| | LSH | 12.95 | 71.43 | 61.47 | 55.19 | 2444.05 |
| | None (Base) | 12.41 | 68.20 | 59.13 | 54.01 | 4996.64 |
| Motion Blur | $k$-extension | 14.03 | 65.13 | 53.77 | 48.01 | 2588.06 |
| | thresholds | 13.81 | 68.20 | 57.51 | 51.05 | 4073.04 |
| | LSH | 12.39 | 68.03 | 59.30 | 54.04 | 4030.52 |

Table 11: Performance under visual corruption of our methods applied *independently* to the HAMT agent on the R2R benchmark.

real-world VLN systems. We apply our methods to the HAMT agent and report results on the R2R benchmark in Table 11.

Our methods appear more robust to Low Lighting than Motion Blur, which corroborates our findings in Sec 5.3. Across both corruptions, $k$-extension and early-exiting see a slight increase of 150–200 GFLOPs compared to the results in Table 10. This can likely be attributed to the increased trajectory length, and for early-exiting, we also find that the OOD samples require more ViT layers before sufficiently saturating. Both mechanisms result in significant drops in performance, though less than when we apply all simultaneously (results shown in Table 6). Early-exiting is slightly more robust, achieving a 2–7% higher SR, which makes sense as it processes strictly more images than $k$-extension.

Interestingly, LSH functions extremely well when Low Lighting is applied. It offers a $\sim$49% reduction in GFLOPs, compared to just 18% when no corruption is present. We hypothesize that the reduced lighting makes more images similar, causing our algorithm to find more matches and reuse more embeddings. It also offers significant robustness, only incurring a 1% point drop in SR. It seems like our caching mechanism is better suited for this environment, a finding we hope to explore in future work. For Motion Blur, LSH is less successful, being more robust than our other mechanisms but with minimal computational savings.

## F RELATED WORK ON MODEL COMPRESSION

Research has proposed an orthogonal approach to reduce the computational demands and memory footprint of deep-learning models: *model compression*. Quantization and pruning are the leading practice in model compression. Quantization (Jacob et al., 2018; Choi et al., 2018; Louizos et al., 2018; Bhalgat et al., 2020; Uhlich et al., 2019; Banner et al., 2019; Choukroun et al., 2019; Li et al., 2021; Nagel et al., 2020) transforms the memory representation of model parameters from 32-bit floating point numbers to a lower-bit integers (e.g., 4-bit integers), thereby making it more storage efficient and lowering memory usage. Pruning (Molchanov et al., 2016; Fan et al., 2019; Fang et al., 2023; Nova et al., 2023; Han et al., 2015b;a; Hoang & Liu, 2023) aims to create sparse models by removing parameters that are less important for maintaining performance, effectively reducing model size and computation.

While quantization and pruning have been demonstrated in simpler unimodal encoder settings for image and text, they are much more challenging in vision-language model(VLM) settings (Wang et al., 2022; Sun et al., 2024) and largely unexplored in VLN. (Wang et al., 2022) highlighted the challenges of pruning VLMs due to the unequal weighting of visual and linguistic modalities. They mitigated this by using a modal-adaptive approach, adjusting pruning ratios across different model components based on downstream task sensitivity. Similarly, (Sun et al., 2024) demonstrated that naively applying post-training quantization to CLIP caused significant performance degradation, which they addressed by introducing prompt tuning and alignment modules.

We expect similar challenges to be exhibited by VLN agents, if not exacerbated. VLN models, in addition to processing language and visual modalities, involve sequential decision-making dependent

on actions taken at each time step. We anticipate the complex interactions between these information sources to require careful consideration while adapting model compression techniques. Future research on such techniques can be superposed along with our input-adaptive inference method to develop highly efficient models with an acceptable performance trade-off.

## G  COMPARISON OF ADDITIONAL SIMILARITY METRICS

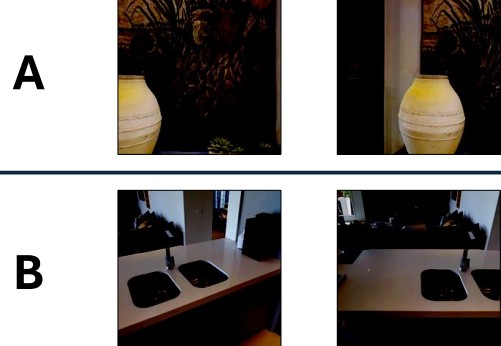

Figure 9: **Two sets of example views (A and B)** demonstrating non-identical but similar views that have been slightly shifted during navigation.

| Simiarlity Metrics | Set A | Set B |
|---|---|---|
| **SSIM** (Wang et al., 2004) | 0.24 | 0.32 |
| **FSIM** (Zhang et al., 2011) | 0.26 | 0.27 |
| **LPIPS** (Zhang et al., 2018) | 0.55 | 0.62 |
| **SURF** (Bay et al., 2006) | 0.31 | 0.32 |
| **SIFT** (Lowe, 2004) | 0.45 | 0.37 |
| **ORB** (Rublee et al., 2011) | 0.07 | 0.19 |

Figure 10: **Similarity scores measured on Set A and B.** We test 6 different similarity metrics.

Other than the three similarity metrics we use, we test three additional metrics for comparison: SURF (Bay et al., 2006), SIFT (Lowe, 2004), and ORB (Rublee et al., 2011). These are feature detection and description algorithms designed to identify and match keypoints in images. The similarity scores are computed by dividing the number of matching keypoints by the minimum number of keypoints detected in the two images. We test all six algorithms on two sets of scenes, reflecting shifts caused by an agent's changing perspectives during navigation.

Figure 9 illustrates the two sets of scenes, and Table 10 summarizes the quantitative comparison. Among the three metrics we employ for our main evaluation, LPIPS demonstrates a higher similarity measure of approximately 60% for both sets. In contrast, SSIM and FSIM are less effective at capturing the similarity between views in Sets A and B. The three additional metrics (SURF, SIFT, and ORB) are also ineffective in providing reliable similarity scores for both image sets A and B. Our qualitative comparison of different similarity metrics applied to sets of similar scenes highlight the challenges these metrics face in accurately identifying true visual similarity. We believe that an accurate measure of scene similarity is crucial for further reducing the computational demands of a VLN agent, and we leave this for future work.

## H  ANALYZING PERFORMANCE-EFFICIENCY TRADE-OFF IN OUR METHOD

In order to illustrate our tunable performance-efficiency trade-off, we show that even when limiting the performance drop to under 5%, our input adaptive inference method applied to the HAMT agent achieves significant computational savings. For reference, the baseline HAMT model achieves a SR of 66.16 with a computational cost of 4763.24 GFLOPs. Figure 11 shows that with a 3–5% drop in SR, we still manage to achieve 43–50% savings in GFLOPs. These results were tested on the R2R validation unseen dataset.

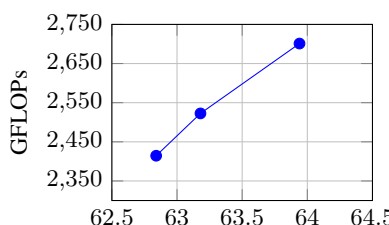

Figure 11: Trade-off between Performance (SR) and GFLOPs

