# OpenReview forum: "Harnessing Input-adaptive Inference for Efficient Vision-and-Language Navigation"
_ICLR.cc/2025/Conference — Submitted to ICLR 2025_

### Official Review · Reviewer_vmJ4 · 2024-11-01

**Soundness:** 3
**Presentation:** 2
**Contribution:** 2
**Rating:** 3
**Confidence:** 4

**Summary:**

This paper proposes an input-adaptive navigation method for efficient vision-and-language navigation (VLN) that addresses the overthinking problem common in history-aware multi-modal transformer models. The authors develop three adaptive mechanisms to improve spatial, model-level, and temporal efficiency.

**Strengths:**

The motivation of this work is strong, as efficiency is indeed a critical issue to address in VLN.

**Weaknesses:**

1. Although the paper frequently emphasizes efficiency and mentions potential deployment on robots, it only provides GFLOPs as a metric. This is insufficient; additional metrics like model size and inference time should be included.
2. The approach reduces computation by limiting the number of panoramic views, but this may not address an actual issue. VLN image features are typically pre-extracted offline, eliminating the need for recalculating visual features during navigation. Moreover, recent methods often select candidate views rather than processing all 36 panoramic views, making the proposed solution potentially less relevant.
3. The authors appear to lack familiarity with VLN, as related work only covers pre-2018 studies, and the experiments compare only with HAMT (NeurIPS 2021) and DUET (CVPR 2023).
4. While GFLOPs are reduced by half, the performance degradation is significant, which may make the results difficult to justify as acceptable.

**Questions:**

See weakness.

---

> ### Author Response · Authors · 2024-11-25
> **Response to Reviewer vmj4**
>
> We thank the reviewer for their time. However, we find several points made to be factually incorrect or subjective. Below, we address the questions and concerns.
>
> ---
>
> > (Weakness 1) Although the paper frequently emphasizes efficiency and mentions potential deployment on robots, it only provides GFLOPs as a metric. This is insufficient; additional metrics like model size and inference time should be included.
>
> We argue that GFLOPs is a sufficient metric for measuring efficiency, particularly model-level speed-up. The number of GFLOPs needed to process an input is proportional to the inference time. We choose it over inference time because (1) it is agnostic to the hardware configuration used for evaluation and (2) it is less prone to measurement errors. To contextualize our performance with respect to wall-clock time, we provide further results in Table 8 in Appendix D. Finally, adding model size as an efficiency metric is inappropriate in this context since our methods do not alter parameter count. We point the reviewer to more relevant studies on this topic [1], [2], which we consider orthogonal to our work.
>
> [1] EfficientVLM: Fast and Accurate Vision-Language Models via Knowledge Distillation and Modal-Adaptive Pruning. Wang et al., arXiv Preprint 2023.
>
> [2] P4Q: Learning to Prompt for Quantization in Visual-language Models. Sun et al., arXiv Preprint 2024.
>
> ---
>
> > (Weakness 2) VLN image features are typically pre-extracted offline, eliminating the need for recalculating visual features during navigation.
>
> This is not possible in a practical setting. It would be intractable for a VLN agent to pre-extract every possible image it may encounter in deployment; pre-extraction is rather an optimization tool to speed up evaluation within static datasets of simulation environments. Our proposed method offers computational savings in practical settings where pre-extraction is not an option and images must be processed.
>
> ---
>
> > (Weakness 3) Recent methods often select candidate views rather than processing all 36 panoramic views, making the proposed solution potentially less relevant.
>
> This is not correct. While candidate views form the agent's decision space, the agent’s visual encoder still processes all the views in today’s typical agents [1], [2], [3], [4], [5]. They are not ignored and rather convey potentially useful spatial and semantic priors relevant to navigation. Our proposed solution leverages the insight that not all non-navigable (non-candidate) views provide the same amount of relevant information, allowing us to dynamically allocate less compute to them while mitigating performance degradation.
>
> [1] VLN BERT: A Recurrent Vision-and-Language BERT for Navigation. Hong et al., CVPR 2021.
>
> [2] History-Aware Multimodal Transformer for Vision-and-Language Navigation. Chen et al., NeurIPS 2021.
>
> [3] Think Global, Act Local: Dual-Scale Graph Transformer for Vision-and-Language Navigation. Chen et al., CVPR 2022.
>
> [4] Scaling Data Generation in Vision-and-Language Navigation. Wang et al., ICCV 2023.
>
> [5] A New Path: Scaling Vision-and-Language Navigation With Synthetic Instructions and Imitation Learning. Kamath et al., CVPR 2023.

---

> ### Author Response · Authors · 2024-11-25
> **Response to Reviewer vmj4 (cont)**
>
> > (Weakness 4) The authors appear to lack familiarity with VLN, as related work only covers pre-2018 studies, and the experiments compare only with HAMT (NeurIPS 2021) and DUET (CVPR 2023).
>
> The studies in the related work and models considered in our evaluation were carefully selected and not because we “lack familiarity with VLN.” Here, we justify our decisions regarding both.
>
> **Related Work**
>
> Our related work includes several more recent (post-2018) studies [1], [2], [3]. Our aim in this section was to provide work similar to our research (particularly the HAMT [2] and DUET [3] agents) rather than a complete overview of the VLN field. However, we acknowledge that the related work section would benefit from including more recent research, e.g., newer discrete VLN agents like ScaleVLN [4] (which we originally cited outside of the related work) and MARVAL [5], and will update the section accordingly.
>
> **Models Used in the Experimental Evaluation**
>
> We believe our selection of the HAMT and DUET agents enabled a representative evaluation. These agents are frequently used in other works, and DUET is the baseline architecture of the current state-of-the-art ScaleVLN [4]. Provided that most recent VLN agents [1], [2], [3], [4], [5] leverage similar Transformer-based architectures, we reason that our proposed method is largely generalizable. This is particularly true because the usage of a ViT for image encoding is consistent across all modern VLN agents, which we identify as the computational bottleneck in Section 4.1. All of our proposed techniques reduce the computations of this component, and there is no reason to believe that transferring them to other agents would be less effective. Our results support this claim. HAMT and DUET have substantially different architectures, but we show in the main evaluation (Section 5.1 and supplemented in Appendix D) that we achieve considerable computational savings for both.
>
> [1] VLN BERT: A Recurrent Vision-and-Language BERT for Navigation. Hong et al., CVPR 2021.
>
> [2] History-Aware Multimodal Transformer for Vision-and-Language Navigation. Chen et al., NeurIPS 2021.
>
> [3] Think Global, Act Local: Dual-Scale Graph Transformer for Vision-and-Language Navigation. Chen et al., CVPR 2022.
>
> [4] Scaling Data Generation in Vision-and-Language Navigation. Wang et al., ICCV 2023.
>
> [5] A New Path: Scaling Vision-and-Language Navigation With Synthetic Instructions and Imitation Learning. Kamath et al., CVPR 2023.
>
> ---
>
> > (Weakness 5) While GFLOPs are reduced by half, the performance degradation is significant, which may make the results difficult to justify as acceptable.
>
> We find this remark highly subjective. In our main evaluation (Section 5.1), we report an approximate 60% computational savings (2.5x speed up) with a 10% reduction in performance (SR) across most tasks. This performance drop is comparable to existing work in the input-adaptive domain [2], [3]. Furthermore, results from Table 2 indicate that we achieve ~61 SR with HAMT while providing over a 2x reduction in computations. This is a respectable performance and would have been close to SOTA (RecBERT [1]) until HAMT in 2021.
>
> More importantly, our contribution is not a fixed computational speed-up at a fixed performance degradation rate but rather a tunable tradeoff between computational and accuracy requirements. For example, in Table 3, we show that SR can be recovered from ~61–63 while maintaining a ~2x speed up by increasing the number of k-extensions from 4–6. We further discuss this trade-off in Appendix G. Our main evaluation presents the best tradeoff between accuracy and efficiency, but the proposed solution is dynamic by design.
>
> [1] VLN BERT: A Recurrent Vision-and-Language BERT for Navigation. Hong et al., CVPR 2021.
>
> [2] DeeBERT: Dynamic Early Exiting for Accelerating BERT Inference. Xin et al., ACL 2020.
>
> [3] BERT Loses Patience: Fast and Robust Inference With Early Exit. Zhou et al., NeurIPS 2020.

---

> > ### Comment · Reviewer_vmJ4 · 2024-11-25
> >
> > After reviewing the authors’ response, I find that my concerns remain unresolved.
> >
> > First, the authors emphasize that their method is intended for real-world environments, but their experiments are conducted solely in a simulator. The key issue of whether candidate viewpoints' image feature can be pre-extracted hinges on this point. If the authors insist that their method is designed for real-world applications and preprocessing all location images is infeasible, they should provide experimental results on a physical robot or in continuous environments such as R2R-CE. However, I see no such results. Instead, the authors continue to focus on discrete environments while emphasizing the practical applicability of their method, which is contradictory. I strongly encourage the authors to reconsider their response carefully.
> >
> > Second, the authors justify limiting their related work discussion to studies up to 2018 by stating that more recent works are irrelevant to their research. This demonstrates a lack of familiarity with the VLN field. It appears the authors are unaware of recent work in VLN that focuses on efficiency and deployment on physical robots.
> >
> > Regarding the efficiency results, I find them difficult to accept. I urge the authors to consider more recent works, such as MAGIC [1]. For example, MAGIC achieves comparable SR to the original model with only 25.30% of the GFLOPs, and even when reducing GFLOPs to 3.08%, the SR drops by just 4 points.
> >
> > [1] MAGIC: Meta-Ability Guided Interactive Chain-of-Distillation for Effective-and-Efficient Vision-and-Language Navigation. Wang, L., He, Z., Shen, M., Yang, J., Liu, C., & Chen, Q. ArXiv, abs/2406.17960.
> >
> > In conclusion, the authors’ response further highlights the need for them to develop a deeper understanding of VLN and to review more recent work in this field.

---

### Official Review · Reviewer_VoXA · 2024-11-02

**Soundness:** 2
**Presentation:** 3
**Contribution:** 1
**Rating:** 3
**Confidence:** 4

**Summary:**

This paper presents three methods that improves the computational efficiency for vision-and-language navigation (VLN), especially in the context of visual input processing. The goal of proposed methods is selecting a subset of 36 panoramic views while maintaining the overall VLN performance. Experimental results show that the proposed algorithms (early-exit, spatial locality, and temporal locality) improve the computational efficiency in several VLN benchmarks.

**Strengths:**

- The paper addresses the critical problem in VLN when deployed in real-world applications.
- The paper is well-written and easy to understand.

**Weaknesses:**

- The paper highlights the practical aspects of VLN when deployed in real-world environments. Ironically, the proposed methods lack practicality in their current form.
  - The method regarding the spatial locality requires "navigable points" in the agent's current position. However, in real-world environments, no navigable points are given to VLN agents. This indicates that we can't directly apply this method to the real-world VLN scenario. How can we apply the spatial locality method when navigable points are not given? One way to address this concern is to investigate the results by employing the waypoint predictor. In the task of vision-and-language navigation in continuous environments (VLN-CE), the work [3] has trained the waypoint predictor that predicts several navigable points in each time step. It would be interesting to see the comparison when given the estimated navigable points and the ground-truth (i.e., known) navigable points.

  - Another concern is about the temporal locality method. The method benefits from the computational efficiency, but it also sacrifices the space complexity of the VLN system. This can be a critical problem when VLN agents should perform long-horizon navigation. I think one way to address this concern is to perform a quantitative analysis of the space-time tradeoffs, especially for longer navigation tasks. In this case, what should be a metric for the memory usage?


- Experiments were conducted in limited settings. Most of experiments are studied in the R2R task, and two VLN models in the paper (HAMT and DUET) are quite outdated. It request the authors to investigate the effect of three proposed methods in longer-horizon VLN tasks (e.g., RxR) with more recent state-of-the-art models (e.g., ScaleVLN [1] or MARVAL [2]). More specifically,

  - The task length of the R2R task is relatively short, so I have concerns that the performance drop of applying the three proposed methods will become more prominent as the task length increases. Can the authors report how performance and computational efficiency vary with task length? I recommend to investigate the RxR task since this benchmark has longer navigation length. Furthermore, comparing the contribution of each method in longer navigation tasks will be an another interesting analysis since it clearly shows which methods are robust to task length.





References

[1] Scaling data generation in vision-and-language navigation. Wang et al., ICCV 2023.

[2] A new path: Scaling vision-and-language navigation with synthetic instructions and imitation learning. Kamath et al., CVPR 2023.

[3] Bridging the Gap Between Learning in Discrete and Continuous Environments for Vision-and-Language Navigation. Hong et al., CVPR 2022.

**Questions:**

N/A

---

> ### Author Response · Authors · 2024-11-25
> **Response to Reviewer VoXA**
>
> We thank the reviewer for their time and valuable feedback. Below, we provide answers to the questions and concerns. We will include this discussion in the final version of our paper.
>
> ---
>
> > (Weakness 1) The proposed methods lack practicality in real-world environments.
>
> We appreciate the reviewer's concern with the practicality of our techniques in real-world environments. Here, we address each primary point made by the reviewer individually.
>
> **No Navigable Points are Given to VLN Agents in Real-World Environments**
>
> The lack of navigable viewpoints in the real world indeed prevents the immediate deployment of our proposed technique. This is a substantial challenge to VLN as a whole. As the reviewer suggests, using our spatial locality methods in continuous/real-world environments would require predicting navigable views, for which prior work provides several demonstrations [1], [2], [3], [4]. Assuming we do not modify this prediction process, we suspect our proposed techniques would remain effective as the insights of spatial locality still apply. This is reasonable when considering the advancements of these waypoint prediction methods, the most recent of which [4] achieves a considerable SR of 57 on the R2R-CE benchmark (with the closest work [3] achieving an SR of 44).
>
> Unfortunately, incompatibility between discrete VLN benchmarks and existing waypoint prediction methods and the non-trivial task of transferring discrete agents to continuous environments makes this hypothesis challenging to empirically evaluate. We leave the study of efficient VLN in continuous environments as future work and thank the reviewer for raising this issue.
>
> [1] Sim-to-Real Transfer for Vision-and-Language Navigation. Anderson et al., CoRL 2020.
>
> [2] Sim-2-Sim Transfer for Vision-and-Language Navigation in Continuous Environments. Krantz and Lee, ECCV 2022.
>
> [3] Bridging the Gap Between Learning in Discrete and Continuous Environments for Vision-and-Language Navigation. Hong et al., CVPR 2022.
>
> [4] ETPNav: Evolving Topological Planning for Vision-Language Navigation in Continuous Environments. An et al., TPAMI 2024.
>
> **Space Complexity of Locality Sensitive Hashing**
>
> The reviewer is correct to point out that our locality-sensitive hashing (LSH) technique incurs a storage overhead, which we regrettably do not highlight in our current evaluation. Here, we provide some information regarding this overhead and how it impacts the deployment of the technique in practice.
>
> The worst-case scenario for LSH is that the agent caches every image it encounters on a long navigation route. In our experiments, we approximate the overhead of this scenario to total approximately 522.7 MB. Specifically, our LSH technique stores pairs of images and embeddings. In the benchmarks we consider, these images are of size 3x224x224. The embedding size depends on the model, which for HAMT and DUET is 197x768 (the number of ViT patches times the model’s hidden dimension). These are stored in full-precision floating-point format (32 bits per value), resulting in (3 * 224 * 224 + 197 * 768) * 32 bits of storage per cached pair, approximately 1.2 MB. In our experiments, the longest navigation route was 12 steps (from R2R-Back), and if we assume all 36 images per panorama are cached, we obtain the worst-case overhead. In practice, however, we find that most tasks are 5–7 steps, and we cache roughly 14 images per step, producing a more typical overhead of 84.7–118.6 MB. Considering that modern VLN agents are orders of magnitude larger, this is not a limiting factor to practical deployment.
>
> Additionally, our mechanism is easily adaptable to storage overhead requirements. By limiting the number of cached view-embedding pairs, one can ensure that the storage overhead does not exceed a set threshold. In our work, we consider the best-case functioning of the LSH algorithm such that an upper bound can be identified and compared to our other techniques. We leave the exploration of storage-efficient hashing mechanisms as future work.
>
> Placing this discussion in the context of our entire approach, the LSH mechanism provides a noticeable but substantially smaller speed-up compared to k-extensions and early-exiting (~2%). If needed, an aggressive storage overhead limit can be placed without significantly impacting the computational savings of our method. We thank the reviewer for highlighting this limitation and will include it in our final paper.

---

> ### Author Response · Authors · 2024-11-25
> **Response to Reviewer VoXA (cont)**
>
> > (Weakness 2) The experiments were conducted in limited settings.
>
> We appreciate the reviewer’s concern. Here, we first explain our reasoning for using the VLN agents HAMT [1] and DUET [2] and then discuss the efficacy of our method on different benchmarks with varying path lengths.
>
> **Usage of HAMT and DUET in the Evaluation**
>
> We selected the HAMT [1] and DUET [2] agents to evaluate our proposed method because they are largely representative works. These agents are frequently used in recent work, and DUET is the baseline architecture of the current state-of-the-art ScaleVLN [3]. The primary difference between DUET and ScaleVLN is that ScaleVLN uses scaled training environments. There is no reason to suspect data augmentation has a substantially negative effect when we apply our speed-up techniques. We also appreciate the reviewer’s suggestion of using MARVAL [4], but unfortunately, to our knowledge, it is not publicly available.
>
> As all recent VLN agents [1], [2], [3], [4], [5] leverage similar Transformer architectures, particularly a computationally expensive ViT for image processing, we reason that the proposed method is largely generalizable. Our main evaluation (Section 5.1) supports this claim, where we achieve substantial computational efficiency on both HAMT and DUET despite their architectural differences.
>
> **Evaluating Other Benchmarks and the Robustness of Our Method to Navigation Length**
>
> We first note that we complement the main experimental results on R2R (Table 2 in Section 5.1) with several other benchmarks (R2R-Back, R2R-Last, REVERIE, CVDN, and SOON) in Appendix D (results in Table 7). However, we originally did not consider the robustness of our proposed techniques to path length, and we thank the reviewer for highlighting it. Here, we study if the errors introduced by our techniques propagate to longer path lengths. For several benchmarks, we report the average path length (measured as the minimal number of navigation actions needed to reach the target destination), change in navigation error (average distance of the agent’s final position to the target position), and change in GFLOPs, compared to the base model. Results are shown in the table below.
>
> | Agent | Benchmark | Average Path Length | Change in Navigation Error | Change in GFLOPs |
> |-------|:-----------:|:---------------------:|:----------------------------:|:------------------:|
> | HAMT  | R2R   	| 6.0             	| +0.53                  	| -2845.63     	|
> |   	| R2R-Last  | 6.0             	| +0.45                  	| -2393.24     	|
> |   	| R2R-Back  | 12.0            	| +0.54                  	| -5463.98     	|
> | DUET  | R2R   	| 6.0             	| +0.68                  	| -2971.7      	|
> |   	| SOON  	| 9.6             	| -0.44                  	| -5463.98     	|
>
> We find our method is robust to longer path lengths. The change in navigation error does not increase, and we even see a decrease for the SOON benchmark, which has a path length noticeably longer than R2R. The results also show that for longer paths, our efficient VLN agent sees roughly proportional computational savings. For example, the average path length in R2R-Back is double R2R, and we achieve a 1.92x larger reduction in GFLOPs for the HAMT agent.
>
> We agree with the reviewer that a deeper analysis of each proposed technique's robustness to path length would be insightful. Unfortunately, we are unable to address this within the rebuttal period. However, given our results, we do not suspect substantial discrepancies between the individual techniques and leave further investigation as future work.
>
> We hope these results address the reviewer’s concerns, and we will update our final paper to include this valuable discussion.
>
> [1] History Aware Multimodal Transformer for Vision-and-Language Navigation. Chen et al., NeurIPS 2021.
>
> [2] Think Global, Act Local: Dual-Scale Graph Transformer for Vision-And-Language Navigation. Chen et al., CVPR 2022.
>
> [3] Scaling Data Generation in Vision-and-Language Navigation. Wang et al., ICCV 2023.
>
> [4] A New Path: Scaling Vision-and-Language Navigation With Synthetic Instructions and Imitation Learning. Kamath et al., CVPR 2023.
>
> [5] VLN BERT: A Recurrent Vision-and-Language BERT for Navigation. Hong et al., CVPR 2021.

---

> > ### Comment · Reviewer_VoXA · 2024-12-03
> > **Official Comment by Reviewer VoXA**
> >
> > I would like to thank the authors for devoting their time to address my concerns.
> >
> > I carefully read the author's response, as well as the rebuttal with other reviewers. Unfortunately, the response does not fully address my concerns, as it relies on an educated guess (no navigable points) or preliminary analyses (e.g., space-time tradeoffs and robustness analysis). I believe that I am not ready to improve my ratings at this time.

---

### Official Review · Reviewer_xson · 2024-11-03

**Soundness:** 3
**Presentation:** 3
**Contribution:** 3
**Rating:** 6
**Confidence:** 4

**Summary:**

This paper proposes an efficient input-adaptive inference method for vision-and-language navigation (VLN) to address computational “overthinking,” where models perform excessive calculations unnecessarily. The approach combines three adaptive strategies: selective processing of critical views (spatial), dynamic thresholding for early exits (model-level), and caching of repeated views (temporal). Evaluated on six VLN benchmarks, this method achieves substantial computational savings—up to 60%—with minimal performance loss, offering a practical solution for improving VLN efficiency in resource-constrained settings.

**Strengths:**

1. The paper introduces an innovative approach to enhancing efficiency in vision-and-language navigation (VLN) by addressing "overthinking"—the unnecessary computation in model decision-making. The authors propose a unique combination of adaptive methods across spatial, model-level, and temporal dimensions. This combination is well-suited to VLN, where inputs are sequential and often contain redundancies.
2. The method is well-designed, clearly breaking down the adaptive components into three strategies: spatial locality, model-level efficiency, and temporal locality. The paper is generally well-written.
3. The proposed method has practical implications for VLN tasks in resource-constrained environments, such as robotic navigation. By demonstrating a 60% reduction in computation with minimal impact on performance, the work provides a valuable solution for deploying VLN agents in real-world scenarios where computational resources are limited.

**Weaknesses:**

1. While the paper effectively introduces adaptive mechanisms (spatial, model-level, and temporal), there is limited theoretical analysis of each mechanism's impact on VLN performance and generalization. For example, the paper could explore how the choice of k (in the k-extension) or the aggressiveness of the adaptive thresholding influences model stability and long-term performance in complex environments.
2. Some formulae and symbols, such as those in the k-extension and thresholding sections, could benefit from additional clarification and consistent notation to improve readability. For instance, consistently defining parameters and variables at the start of each section would reduce potential confusion.
3. The paper provides a robustness analysis under various visual corruptions, but it treats the proposed method as a single unit. A more detailed breakdown of how each mechanism (spatial, model-level, temporal) contributes to robustness under different corruptions (e.g., which mechanism is most affected by motion blur or low lighting) would offer valuable insights.
4. The caching mechanism, which leverages locality-sensitive hashing (LSH) to avoid recomputing similar views, is a valuable addition. However, the paper does not fully explore how this caching performs under different levels of scene variability or how effective it is when environmental conditions shift significantly (e.g., new objects or layouts). A more detailed sensitivity analysis on the caching's impact across different types of VLN tasks or environmental settings would help establish a clearer understanding of when this mechanism is most beneficial.

**Questions:**

Please refer to the weaknesses section.

---

> ### Author Response · Authors · 2024-11-25
> **Response to Reviewer xson**
>
> We thank the reviewer for their time and valuable feedback. Below, we provide answers to the questions and concerns. We will include this discussion in the final version of our paper.
>
> ---
>
> > (Weakness 1) There is limited theoretical analysis of each mechanism's impact on VLN performance and generalization.
>
> We agree that independently analyzing the effectiveness of each proposed technique is very important. In Section 5.2, we provide an initial discussion by exploring the impact of k (the number of extended non-navigable views) and the early exit thresholds on navigation success. We also consider how using different similarity metrics for hashing images impacts our locality-sensitive hashing (LSH) technique (with further discussion and results in Appendix F).
>
> To address the reviewer's concerns further, we report the results of deploying each mechanism (k-extensions, early-exiting, and LSH) independently. Below, we show the same metrics detailed in Section 3 for the HAMT agent on the R2R benchmark.
>
> | Method                 |   TL  |  OSR  |   SR  |  SPL  |  GFLOPs |
> |------------------------|:-----:|:-----:|:-----:|:-----:|:-------:|
> | k-extension            | 12.52 | 71.86 | 61.30 | 55.79 | 2408.99 |
> | thresholds             |   12.33   |   72.46   |  62.62   |   57.39   |  3867.46  |
> | LSH                    |   11.53   |   74.20   |   66.11   |   61.47   |    3,894.76    |
>
> The results show that k-extension offers substantially more speed-up than early-exiting (thresholds) and LSH. This makes sense, as zeroing out even a few views will be more efficient than processing them through a fraction of the ViT’s layers, and LSH only hashes and reuses a small subset of views. It also only degrades performance slightly more than early-exiting, validating our insight that views spatially distant from the navigable views are not as important for predictions. LSH provides the best performance among individual mechanisms, suggesting the reused embeddings are near-identical. We have added these results and discussion to Appendix E.
>
> We hope this discussion addresses the reviewer’s concerns and are happy to answer any further comments/questions.
>
> ---
>
> > (Weakness 2) Some formulae and symbols, such as those in the k-extension and thresholding sections, could benefit from additional clarification and consistent notation to improve readability.
>
> Thank you for pointing this out; we agree clarity can be improved. In the final version of our paper, we will define and introduce formulae and symbols more consistently and clearly.
>
> ---
>
> > (Weakness 3) The paper provides a robustness analysis under various visual corruptions, but it treats the proposed method as a single unit.
>
> We thank the reviewer for pointing out this limitation. Analyzing the impact of visual corruption on each proposed technique would provide valuable insights regarding its robustness. Please refer to the follow-up comment where we address this concern.
>
> ---
>
> > (Weakness 4) The paper does not fully explore how locality-sensitive caching (LSH) performs under different levels of scene variability or how effective it is when environmental conditions shift significantly.
>
> This is a valid concern, and exploring the performance of LSH to scene variability is an interesting direction. We provide some preliminary results on this in Appendix F, showing how the slight translation of an image has a large effect on even the most recent and performant image similarity metrics. This suggests that including new objects or layouts would likely have a substantial impact on the performance of our caching algorithm.
>
> Unfortunately, further analyses in this direction are challenging. Existing VLN benchmarks are static, so the agent will always encounter the same scene at a given step. This limits our ability to test scene variability, particularly adding/removing objects or changing their orientation. To adequately address this concern, which may have consequences on general VLN agent performance, it would likely require the development of new datasets. We leave this intriguing implication as future work and thank the reviewer for highlighting it.

---

> ### Author Response · Authors · 2024-11-28
> **Response to Reviewer xson (update)**
>
> **(Follow-up comment)** We provide a per-mechanism robustness analysis by running all techniques individually on the “Motion Blur” and “Low Lighting” corruptions, chosen based on their varying impact on performance (detailed in Section 5.3) and the likelihood of occurring in real-world VLN systems. We use the same metrics from Section 3 and report results on the HAMT agent on the R2R benchmark.
>
> |  Corruption  | Method        |   TL  |  OSR  |   SR  |  SPL  |  GFLOPs |
> |:------------:|---------------|:-----:|:-----:|:-----:|:-----:|:-------:|
> | Low Lighting | None (Base)         | 12.15      |  71.31     | 62.58      |  57.23     |   4903.06      |
> |              | k-extension | 13.86 | 71.14 | 57.34 | 50.78 | 2571.06 |
> |              | thresholds    | 13.63 | 70.29 | 58.79 | 52.16 | 4099.21 |
> |              | LSH           | 12.95 | 71.43 | 61.47 | 55.19 | 2444.05 |
> |  Motion Blur | None (Base)         | 12.41      |  68.20     | 59.13      | 54.01      | 4996.64        |
> |              | k-extension | 14.03 | 65.13 | 53.77 | 48.01 | 2588.06 |
> |              | thresholds    | 13.81 | 68.20 | 57.51 | 51.05 | 4073.04 |
> |              | LSH           | 12.39 | 68.03 | 59.30 | 54.04 | 4030.52 |
>
> Our individual mechanisms appear more robust to Low Lighting than Motion Blur, which corroborates our findings
> in Sec 5.3. Early-exiting appears slightly more robust than k-extension, achieving a 2–7% higher SR, which makes sense as it processes strictly more images. Interestingly, LSH functions extremely well when Low Lighting is applied. It offers a 49% reduction
> in GFLOPs with only a 1% point drop in SR. We hypothesize that the reduced
> lighting makes more images similar, causing our algorithm to find more matches and reuse more
> embeddings. For Motion Blur, LSH is less successful, being more robust than our other mechanisms but
> with minimal computational savings. For our full results and discussion, please see Appendix E.
>
> We thank the reviewer for suggesting this insightful analysis and hope our results address any remaining concerns.

---

### Official Review · Reviewer_Xs1i · 2024-11-09

**Soundness:** 3
**Presentation:** 2
**Contribution:** 2
**Rating:** 5
**Confidence:** 4

**Summary:**

The work proposes a mixture of techniques leveraging spatial significance and temporal continuity as well as an adaptation of the MuE early exit protocol to alleviate the compute burden of VLN models. The authors show an average reduction by over half of GFLOPs to run the models with drops in success rate performance within 10% of the full model.

**Strengths:**

The paper is well motivated and shows how providing models with better data design, i.e. doing some signal extraction amongst the input noise, can substantially help model efficiency.

The main strength of the work is its clarity in the design of solutions.
- Each of the spatial consideration, temporal hashing and adaptive early exit thresholding are intuitive and simple.
- The combination of the three ingredients is straightforward (for the most part see questions)

The paper's numerical results are comprehensive and consider a variety of relevant benchmarks, showing the merit of the technique. The numerical results obtained are good and proving not only the soundness but also the impact of adaptive input data curation.

**Weaknesses:**

The paper tackles a specific setting, that of VLNs, but more specifically a data pipeline that consists of panorama views (36 precisely) and focuses on simple ways to filter out unnecessary noise in the data and reduce superfluous ViT computations in the image encoder.
- This choice of problem is in my view a bit restrictive, although VLNs are popular models and have attracted a lot of work and attention, they are constantly evolving. Hence focusing on the 36 image panorama as well limiting the application to ViT encoders leaves out room for applications to variants (depth or multimodal sensing, state space models etc...)
- Given the limited applicability context, one would expect a highly specified solution, yet the protocol proposed is quite simple and feels like it could be refined further (additional work on customizing the adaptive MuE perhaps, more specific selection of neighboring views with higher useful information content and so on)

In other words, the setting is not large enough for simple techniques to prove their merit in terms of universality and practicality. Also, the techniques are not tailored enough to the specific setting to maximize the potential gains of the approach and achieve maximal efficiency and performance.

It is unclear what the significance of the result is in practice. The authors express their hope that their "results will inspire future research on developing efficient navigation methods and their deployment in real-world VLN settings."
The gain of a factor of 2 in GFLOPs is not a gain of orders of magnitude and some VLN models are already running on board larger hardware in the real world. The work does not reduce the size of the models (compression) or provide gains large enough to enable such large models to run on edge devices.

Along these lines the paper would be much improved if the authors showed real-world experiments, where in practice the 2x gain offers behavior gains that are desirable from the human user's point of view.

At the clarity level, although the paper is globally well written and easy to follow, some concepts and parts of the protocol are brushed over and require more precision (see questions). Also, a big gap is a per technique analysis, i.e. what how much pulling relies on masking non useful frames vs time similarity hashing vs adaptive MuE as well as combinations (3 individual, and 3 combinations of 2 out of 3 techniques to compare to the 3 simultaneously). Slight adjustments to the techniques should allow a meaningful comparison along these lines.

**Questions:**

- What is a navigable view? This concept central to the filtering of frames as well as the structure of the solution is repeatedly used without being rigorously introduced.
     - How are navigable views identified amongst the 36?

- Lines 201-209 offer a very vague justification of the reasons behind the failure of vanilla MuE. The authors just acknowledge differences in behavior after a couple of steps and conclude "Processing fewer transformer layers can lead to an inaccurate understanding of the visual surroundings. As shown in the bottom-right figures, while the bathroom is consistently visible across multiple steps (t P r2, 10s) in the panorama, the MuE agent fails to recognize it and makes sub-optimal decisions at each navigation step."
     - Why does it fail to recognize it? which views are quitting early? the useful ones ? With only 36 views one would be able to maybe look into the model to get more insight

- How does the budgeted batch inference (introduced line 284) work? It might be a method from a cited paper, but it seems like a crucial underlying piece of the puzzle and might require additional explanation.

- Algorithm 1 shows that navigable views undergo a separate treatment from proximity views. If I understand this correctly navigable views go through the ViT with no early exit or time similarity checks. There is thus no compute gain for these and the gains are only made on masked views (spatial, no compute at all) and proximity views (both hashing and early exit). Can the authors confirm that that is indeed the case?
    - Whether affirmative or not this would require, at least at my level of understanding, a bit clarification in the main text as the algorithm notes are not elucidating enough regarding these crucial aspects of the solution.

---

> ### Author Response · Authors · 2024-11-25
> **Response to Reviewer Xs1i**
>
> We thank the reviewer for their time and valuable feedback. Below, we provide answers to the concerns and questions. We will make sure to incorporate our answers into the final version of our paper.
>
> ---
>
> > (Weakness 1) The choice of the problem is restrictive (specifically, 36-view panoramas and usage of ViT encoders).
>
> We thank the reviewer for this concern. We first point out that the problem we consider, namely, panorama-based visual environments with agents employing ViTs for encoding, has become the predominant setting for VLN. A majority of recent studies utilize these components [1], [2], [3], [4], [5], with prior work showing navigation via panoramas to be practical in real-world demonstrations [6]. Therefore, while it may seem restrictive, it is well-positioned within the current direction of the field.
>
> Importantly, we contribute a set of largely generalizable techniques amongst models currently being developed and deployed. We support this claim by achieving significant computational savings for the HAMT [1] and DUET [2] agents despite their architectural differences. DUET is also the baseline architecture of the current state-of-the-art ScaleVLN [3], the primary difference being scaled training environments, and there is no reason to suspect that such data augmentations would limit our method's transferability. Furthermore, as our most effective contribution, k-extension, only requires the usage of panoramas, it may be applied to the broader usage of architectures in the field.
>
> As with any architecture-oriented optimizations, assuring the applicability of our work to future advancements in the VLN architecture or pipeline is challenging. However, our results show that the proposed method is largely transferable and effective within the current state of the field, making it a practical contribution that may influence future work.
>
> [1] History-Aware Multimodal Transformer for Vision-and-Language Navigation. Chen et al., NeurIPS 2021.
>
> [2] Think Global, Act Local: Dual-Scale Graph Transformer for Vision-And-Language Navigation. Chen et al., CVPR 2022.
>
> [3] Scaling Data Generation in Vision-and-Language Navigation. Wang et al., ICCV 2023.
>
> [4] A New Path: Scaling Vision-and-Language Navigation With Synthetic Instructions and Imitation Learning. Kamath et al., CVPR 2023.
>
> [5] VLN BERT: A Recurrent Vision-and-Language BERT for Navigation. Hong et al., CVPR 2021.
>
> [6] Sim-to-Real Transfer for Vision-and-Language Navigation. Anderson et al., CoRL 2020.
>
> ---
>
> > (Weakness 2) Given the limited applicability context, one would expect a highly specified solution, yet the protocol proposed is quite simple and feels like it could be refined further.
>
> We first highlight that our proposed solution achieves strong, generalizable results that are well within the ideal set by existing input-adaptive work [1], [2], [3]. Regardless of simplicity, this suggests it is a valuable contribution to the field that may enable future work in this direction.
>
> We additionally argue that the proposed solution is specialized. Our spatial locality technique, k-extension, was designed specifically for panorama-based image navigation in which there is a distinction of ‘spatial importance’ relative to navigable views. We also found existing early-exit solutions (i.e., MuE [1]) insufficient, leading us to design one that is more specialized. To our knowledge, our locality-sensitive hashing algorithm is the first such caching technique employed to input-adaptive DNNs, which is only applicable in problem domains where a model encounters repetitive inputs. Broadly speaking, the high-level insights our techniques are built on—leveraging spatial and temporal locality—are largely unique and specialized to the current VLN setting.
>
> Nevertheless, we value the reviewer’s critique and acknowledge the potential for improvement. Further refinement of the early-exiting mechanism and method for selecting views to process are indeed interesting directions that can lead to future solutions. For example, the early-exiting criteria we apply to the ViT (responsible for current observations) is only a function of its activations. However, several other components influence the agent’s navigation (e.g., observation history and language), suggesting there may be alternate criteria that leverage the agent’s entire activation space to determine early exit points. Given the success of the proposed method, we leave these potential explorations as future work and will highlight them in the discussion section of our final paper.
>
> [1] You Need Multiple Exiting: Dynamic Early Exiting for Accelerating Unified Vision Language Model. Tang et al., CVPR 2023.
>
> [2] DeeBERT: Dynamic Early Exiting for Accelerating BERT Inference. Xin et al., ACL 2020.
>
> [3] BERT Loses Patience: Fast and Robust Inference With Early Exit. Zhou et al., NeurIPS 2020.

---

> ### Author Response · Authors · 2024-11-25
> **Response to Reviewer Xs1i (cont)**
>
> > (Weakness 3) The work does not reduce the size of the models (compression) or provide gains large enough to enable such large models to run on edge devices.
>
> The focus of our work is reducing the latency of VLN agents, specifically the time needed to process inputs (visual scene and instruction) and make a decision. While this does not directly enable the deployment of models on edge devices, it is still a large problem in the practical deployment of VLN agents. This latency defines every human-agent interaction and is crucial in establishing the trust and usefulness of these systems. Prior work finds that humans expect a processing latency of no more than 2–4 seconds when interacting with a robot [1]. However, the HAMT agent we study takes ~7 seconds per decision while running on a powerful modern desktop. Clearly, modern VLN agents are far slower than ideal.
>
> However, with the speed-up techniques proposed in this work, the latency associated with ‘overthinking’ can be cut roughly in half (~3.5 seconds in the prior scenario). Our techniques are also composable with orthogonal model efficiency techniques, e.g., pruning and compression, enabling even greater speed-up that approaches human standards. This addresses a key bottleneck in the practical deployment of VLN agents to real-world applications, demonstrating the merit of our work.
>
> [1] Managing delays in human-robot interaction. Pelikan and Hofstetter, ACM Transactions on Computer-Human Interaction, Volume 30, Issue 4, 2023.
>
> ---
>
> > (Weakness 4) Lack of a per-technique analysis.
>
> We agree with the reviewer that a per-technique analysis would provide valuable insights beyond those originally presented in the paper. In Section 5.1, we focus primarily on the impact of k-extensions—finding it offers the best speed-up—and study the benefit of adding early-exiting and locality-sensitive hashing (LSH). This covers most possible combinations (4/7) but, as the reviewer correctly points out, does not elucidate the individual capabilities of each method or combinations of them.
>
> Here, we provide additional results covering the three remaining combinations: early-exiting (thresholds), LSH, and early-exiting with locality-sensitive hashing. We report the same metrics used in our main evaluation (detailed in Section 3) for the HAMT agent on the R2R benchmark in the table below. We include the original k-extension results from Table 2 to compare all 7 combinations of the proposed speed-up techniques.
>
> | Method                 |   TL  |  OSR  |   SR  |  SPL  |  GFLOPs |
> |------------------------|:-----:|:-----:|:-----:|:-----:|:-------:|
> | k-extension            | 12.52 | 71.86 | 61.30 | 55.79 | 2408.99 |
> | thresholds             |   12.33   |   72.46   |  62.62   |   57.39   |  3867.46  |
> | LSH                    |   11.53   |   74.20   |   66.11   |   61.47   |    3,894.76    |
> | k-extension+LSH        | 12.52 | 71.90 | 61.17 | 55.63 | 2013.48 |
> | k-extension+thresholds | 12.89 | 71.95 | 60.41 | 54.57 | 2294.23 |
> | thresholds+LSH            | 12.33 | 72.41 | 62.49 | 57.33 | 3190.66 |
> | All                    | 12.87 | 71.95 | 60.41 |  54.5 | 1917.61 |
>
> Our results show that between individual methods, k-extension offers the most speed-up with a minimal performance drop. This is intuitive, as zeroing out even a few views will be more efficient than processing them through a fraction of the ViT’s layers or hashing a smaller subset. LSH offers the most performance, as the hashed and reused embeddings tend to be near-identical.
>
> The combination we do not present in the original paper, thresholds+LSH, provides slightly better performance than methods using k-extension but at the cost of significantly more compute. This suggests that retaining and partially processing/reusing more non-navigable views mitigates performance drop more than k-extension but is not nearly as efficient. We have added these results and a complete discussion to Appendix E.
>
> We hope these results address the reviewer's concerns and are happy to answer any more comments/questions.

---

> ### Author Response · Authors · 2024-11-25
> **Response to Reviewer Xs1i (cont)**
>
> > (Question 1) What is a navigable view, and how are they identified?
>
> A navigable view is a view within the panorama that the agent can navigate to. It corresponds to locations the agents could move to in their next action, such as a nearby open door or room. As for how they are identified, navigable views are a property of the panorama encountered by the agent and are provided within all benchmarks we evaluate (the standard for discrete VLN). If navigable views are not an inherent property of the environment (as in the real world), prior work has shown they can be accurately predicted [1]. This terminology is consistent with the prior work [2], [3]. We thank the reviewer for this question and will update our final paper to more clearly define the term.
>
> [1] Sim-to-Real Transfer for Vision-and-Language Navigation. Anderson et al., CoRL 2020.
>
> [2] History-Aware Multimodal Transformer for Vision-and-Language Navigation. Chen et al., NeurIPS 2021.
>
> [3] Think Global, Act Local: Dual-Scale Graph Transformer for Vision-and-Language Navigation. Chen et al., CVPR 2022.
>
> > (Question 2) Why does existing input-adaptive work on Transformer encoders (i.e., MuE [1]) fail?
>
> We find that MuE fails in this modality because it underprocesses important views. Standard MuE applies a constant early-exit threshold within all Transformer layers and to all inputs. This results in many inputs exiting at the same layer, even though (as we explore in Section 4.2.1) some are considerably more important than others (i.e., the navigable views). As shown in Figure 3 and discussed in the corresponding Section 4.2, this *one-size-fits-all* approach leads to incorrect selections of navigable views during navigation. To answer the reviewer's questions explicitly, MuE fails to recognize relevant information due to underprocessing both navigable and spatially meaningful (i.e., nearby) non-navigable views.
>
> A natural follow-up question is *why* MuE underprocesses these important views. We explore this in Appendix B due to space limitations but summarize the discussion here. The intuition behind MuE is that the activations of Transformer-based vision models saturate, where their similarity between layers peaks early on and is maintained at future stages of computation. Thus, ideally, later layers introduce negligible new/useful information and can be safely skipped. So, for MuE to be successful, the similarity of activations must sufficiently saturate and not decrease at later layers. However, as shown in Figure 8, the necessary saturation pattern is not observed in the VLN setting. The cosine similarity of activations between layers in HAMT on the R2R task peaks early but then decreases. This explains the significant performance drop when MuE is directly applied to VLN agents, as it consistently exits early despite saturation not being achieved.
>
> We thank the reviewer for posing this question and will modify the final version of our paper to better highlight this discussion.
>
> [1] You Need Multiple Exiting: Dynamic Early Exiting for Accelerating Unified Vision Language Model. Tang et al., CVPR 2023.
>
> ---
>
> > (Question 3) How does the budgeted batch inference work?
>
> Budgeted batch inference [1] is the scenario we consider when designing our speed-up techniques. The premise is that a system must allocate a fixed computational budget amongst several inputs; in the original work, this involved classifying groups of images with uneven compute while retaining high accuracy.
>
> This scenario naturally applies to our work as VLN agents must process several inputs, particularly the individual images comprising the panoramas. However, existing input-adaptive methods on Transformer-based encoders [2] do not work with budgeted batch inference, requiring us to modify the mechanism (see Section 4.2.2 for more details). Budgeted batch inference informed our modifications to the early exit mechanism but is not a crucial component of the algorithm overall.
>
> [1] Multi-Scale Dense Networks for Resource-Efficient Image Classification. Huang et al., ICLR 2018.
>
> [2] You Need Multiple Exiting: Dynamic Early Exiting for Accelerating Unified Vision Language Model. Tang et al., CVPR 2023.
>
> ---
>
> > (Question 4) Which views are processed by which proposed techniques?
>
> The reviewer is correct. We observed substantial performance degradation when attempting to early exit the navigable views and thus opted to fully process them and prioritize our computational savings on the non-navigable views. If a view falls within the extended range defined by k-extension, we apply hashing and early-exit; otherwise, it is not processed. We will make this distinction more explicit in the main text.

---

> > ### Comment · Reviewer_Xs1i · 2024-12-03
> > **Revision post rebuttal**
> >
> > I would like to thank the authors for their clarifications and running insightful per technique quantitative analysis with the limited timeframe. The replies have been helpful in better understanding the work and merit of techniques introduced. I have increased my score from 3 to 5. I am however reluctant to offer the paper a higher score as I maintain my doubts regarding the broader impact of the work as a research manuscript.

---

### Meta-Review · Area_Chair_JS88 · 2024-12-20

**Metareview:**

(a) This paper proposes an input-adaptive navigation method to improve the computational efficiency of Vision-and-Language Navigation (VLN) agents, particularly those based on multi-modal transformer models. The authors argue that existing VLN agents suffer from "overthinking," performing excessive computations on less relevant inputs. To address this, they focus on exploiting the spatio-temporal localities unique to VLN tasks: (1) The spatial locality (reducing the number of navigable views and a few neighboring views that the encoder should process); (2) The temporal locality (reducing compute on identical or nearly identical views in consecutive navigation steps); (3) developing an algorithm for dynamically adapting the thresholds for early-exit.

(b) Strengths:
- Clearly identified problem and motivation: The paper clearly identifies the problem of computational inefficiency in VLN agents and provides a strong motivation for addressing it.
- Novel and well-designed method: The proposed combination of adaptive mechanisms is novel and well-suited to the specific characteristics of VLN tasks, effectively leveraging spatial and temporal redundancies in the input data.
-  Comprehensive evaluation and analysis.

(c) Weaknesses:
- Limited real-world applicability: The reliance on environment-defined "navigable views" limits the applicability of the method in real-world environments where such information is not readily available. Although the authors argued that prior methods have shown that "navigable views" can be predicted in real world in rebuttal, but they didn't present such experiments.
- No real experiments: Despite the emphasis on real-world applicability, the lack of experiments on real robots or in continuous environments weakens the paper's claims.

(d) The decision is to reject the paper, mostly for lack for real-world validation or proofs of transferability.

**Additional Comments On Reviewer Discussion:**

During the rebuttal period, several key points were raised by reviewers:

Reviewer VoXA and vmJ4 expressed concerns about the paper's emphasis on real-world applicability while the experiments were conducted solely in a simulator. This remains an issue during the rebuttal.

Reviewers Xs1i and xson suggested clarifications and additional analysis. Reviewer Xs1i requested a clearer definition of "navigable view" and a more detailed explanation of the failure of vanilla MuE. Reviewer xson suggested a more detailed breakdown of the robustness analysis under different corruptions and a sensitivity analysis of the caching mechanism. The authors responded positively to these requests, providing detailed explanations and promising to incorporate them into the final version of the paper.

The authors were responsive to the reviewers' comments and provided reasonable justifications for their design choices and experimental setup. They also demonstrated a willingness to improve the paper's clarity and include additional analysis. However, concerns regarding the real-world applicability persisted.

While the paper presents an interesting approach to improving the efficiency of VLN agents, I think it can be improved with validation in real-world scenarios.

---

### Decision · Program_Chairs · 2025-01-22

Reject